# The role of sex work laws and stigmas in increasing HIV risks among sex workers

Carrie E. Lyons [1]*, Sheree R. Schwartz[1], Sarah M. Murray [2], Kate Shannon[3], Daouda Diouf[4],
Tampose Mothopeng[5], Seni Kouanda[6], Anato Simplice[7], Abo Kouame[8], Zandile Mnisi[9], Ubald Tamoufe[10],
Nancy Phaswana-Mafuya[11], Bai Cham[12], Fatou M. Drame[4,13], Mamadú Aliu Djaló[14] & Stefan Baral[1]

Globally HIV incidence is slowing, however HIV epidemics among sex workers are stable or increasing in many settings. While laws governing sex work are considered structural determinants of HIV, individual-level data assessing this relationship are limited. In this study, individual-level data are used to assess the relationships of sex work laws and stigmas in increasing HIV risk among female sex workers, and examine the mechanisms by which stigma affects HIV across diverse legal contexts in countries across sub-Saharan Africa. Interviewer-administered socio-behavioral questionnaires and biological testing were conducted with 7259 female sex workers between 2011–2018 across 10 sub-Saharan African countries. These data suggest that increasingly punitive and non-protective laws are associated with prevalent HIV infection and that stigmas and sex work laws may synergistically increase HIV risks. Taken together, these data highlight the fundamental role of evidence-based and human-rights affirming policies towards sex work as part of an effective HIV response.

[1] Center for Public Health and Human Rights, Department of Epidemiology, Johns Hopkins School of Public Health, 615 N Wolfe St, Baltimore, MD 21205, USA. [2] Department of Mental Health, Johns Hopkins School of Public Health, Hampton House 624 N. Broadway 8th Floor, Baltimore, MD 21205, USA. [3] Centre for Gender & Sexual Health Equity, University of British Columbia, 1081 Burrard St, Vancouver, BC, Canada. [4] Enda Santé, Senegal, 56 Cité Comico VDN, B.P, 3370 Dakar, Senegal. [5] People's Matrix Association, Maseru, Lesotho. [6] Institut de Recherche en Sciences de la Santé, Ouagadougou, Burkina Faso, Institut Africain de Santé Publique, 12 BP 199 Ouagadougou, Burkina Faso. [7] ONG Arc-en-Ciel, B.P., 80295 Lomé, Togo. [8] Ministère de la Sante et de l'Hygiène Publique, Abidjan, Côte d'Ivoire. [9] Health Research Department, Strategic Information Division, Ministry of Health, Cooper Centre Office 106, Mbabane, Eswatini. [10] Metabiota. Avenue Mvog-Fouda Ada, Av 1.085, Carrefour Intendance BP, 15939 Yaoundé, Cameroon. [11] DVC Research and Innovation Office, North-West University, Potchefstroom Campus, Private Bag X6001 Potchefstroom, 2520 Potchefstroom, South Africa. [12] Actionaid, Banjul The Gambia, MDI Road, Kanifing South PMB 450, Serrekunda PO Box 725 Banjul, The Gambia. [13] Gaston Berger University, Department of Geography, School of Social Sciences. BP: 234 - Saint-Louis, Nationale 2, route de Ngallèle, St. Louis, Senegal. [14] Enda Santé, Guiné-Bissau. Bairro Santa Luzia, Rua s/n, CP 1041 Bissau, Guinea-Bissau. *email: clyons8@jhu.edu

In 2019, development and scaling up HIV prevention, diagnostic, and treatment strategies collectively have slowed new HIV infections globally, but not to the extent earlier models had predicted[1]. In part, this may be due to stable or growing HIV incidence among marginalized populations including sex workers in many settings. Around the world, ~1 in 10 sex workers is estimated to be living with HIV[2]. Across concentrated and generalized HIV epidemics, female sex workers consistently bear a disproportionate burden of HIV compared with other cisgender women of reproductive age[3]. Across low- and middle-income countries, sex workers have more than a 13 times increased odds of living with HIV compared with other women[3]. Available data of HIV prevalence among sex workers has increased, with a 2018 review finding data points from 101 countries[2]. Incidence data remain limited, but where available, suggest continued challenges in the coverage of effective HIV prevention and treatment interventions for sex workers[2]. Furthermore, emerging evidence suggests that the unmet HIV prevention and treatment needs within sex work significantly contributes to overall HIV transmission even within generalized epidemics[4]. Even in the presence of sustained programs for sex workers, mathematical models predict that 14–38% of all new HIV infections in Benin, Burkina Faso, and Kenya could directly or indirectly be due to sex work over the next 20 years[4–7]. In the absence of dedicated programs, this estimate increases to 58–89%[4–7]. Taken together, these results suggest an urgent need to improve prevention and treatment services for sex workers across HIV epidemics.

Despite expanded access to antiretroviral therapy, sex workers across sub-Saharan Africa continue to have suboptimal HIV prevention and treatment outcomes[8]. Progress in scaling up programs and sustaining coverage of HIV services is undermined by limited assessment of, and efforts to address structural determinants affecting HIV among sex workers[2]. Studies have primarily focused on individual-level biological and behavioral risks for HIV among sex workers with limited examination of higher level structural determinants[2,9]. For example, a systematic review found that fewer than half of epidemiological studies on HIV acquisition and transmission measure structural determinants and this was even less common in studies specifically among sex workers[10]. Consequently, recommendations have been made for increased integration of structural-level factors into HIV research among sex workers[11]. Although more studies assessing structural determinants of HIV among sex workers are emerging, few exist from across sub-Saharan Africa[2].

The legal environment, including laws, enforcement practices, and justice systems, is a key structural determinant of HIV risk for sex workers and is often stated as a focus for the global HIV response[12–15]. Punitive legal environments consistently increase vulnerabilities among sex workers through further pushing sex work into unregulated and unsafe work environments, increasing economic and residential insecurities, and influencing HIV-risk behaviors[14,16,17]. Moreover, laws criminalizing sex work may both increase sex work-related stigmas and potentially contribute to HIV epidemic growth if the sequelae of these laws increases vulnerability and decreases engagement in HIV prevention and treatment services. HIV risks have been characterized in criminalized settings; however, limited opportunities have been available to compare individual health-related outcomes among sex workers across differing legal contexts. Systematic reviews and meta-analyses have leveraged qualitative and quantitative evidence to assess harms associated with sex work policies[18]. And ecological studies of country-level data have observed a relationship between HIV and sex work laws showing a lower HIV prevalence among sex workers in European countries with partial legalization of sex work compared with criminalized settings[19]. However, characterizing the influence of criminalization and punitive policies and stigmas towards sex work across settings with empirical, individual-level data, remains limited[18–21].

Stigmas exist at individual, interpersonal, and structural levels, and represent a process in which an individual is labeled based on some characteristic linked to a stereotype, often resulting in limited opportunities and well-being[22]. Stigma measurement strategies and mitigation interventions have traditionally focused on HIV-related stigmas, though emerging research has focused on stigmas related to sexual behavior among key populations[23,24]. Sexual behavior stigmas can include anticipated, perceived, or enacted stigmas attributable to an individual's sexual behavior, including engagement in sex work[25]. However, the majority of studies measuring stigmas among key populations have focused on stigmas affecting sexual and gender minorities, with only 2% of identified stigma measurement related to sex work[26]. Understanding sex work-related stigmas, especially across different legal contexts has direct implications for the implementation and effectiveness of HIV treatment programs. Where assessed, perceived, anticipated, and enacted stigmas consistently challenge progress along the HIV treatment cascade by limiting engagement in prevention, care, and treatment services[27–30]. Moreover, intersecting stigmas have been associated with prevalent HIV infection and limited uptake of services along the HIV treatment cascade across sub-Saharan Africa[31–33]. Separate from HIV-associated laws, stigma has also been identified as a driver of the HIV pandemic, and its elimination remains one of the three pillars of the UNAIDS plan to achieve zero new HIV infections by 2030[34]. Both the World Health Organization and UNAIDS have recommended increasing efforts to mitigate stigma as critical to an effective HIV response[35,36]. However, understanding the different ways in which stigma potentiates individual-level HIV risks to inform stigma mitigation efforts across different legal contexts remain limited.

In response, this study aims to use individual-level data to characterize the relationship between sex work laws, stigmas, and HIV risks among female sex workers across sub-Saharan Africa. Collectively, findings from these analyses suggest that increasingly punitive and non-protective laws are associated with increased odds of prevalent HIV infection among sex workers. Furthermore, stigmas and sex work laws may operate synergistically in increasing HIV risks, with generally stronger associations between stigmas and HIV in punitive and non-protective settings compared with partially legalized settings. The results suggest that the increased harmful effects of stigmas in more punitive and non-protective legal contexts may be due to increased barriers in the provision or uptake of efficacious HIV prevention and treatment services, or impunity among perpetrators of stigma and lack of recourse for sex workers experiencing health and social stigmas, or more likely, a combination of the two.

## Results

**Study sample characteristics**. A total of 7259 female sex workers are represented in this analysis.

The distribution of women across countries is: Burkina Faso ($n$ = 698; 9.6%), Cameroon ($n$ = 2255; 31.1%), Cote d'Ivoire ($n$ = 466; 6.4%), Guinea-Bissau ($n$ = 567; 7.8%), Lesotho ($n$ = 744; 10.3%), Senegal ($n$ = 758; 10.4%), South Africa ($n$ = 410; 5.6%), Kingdom of eSwatini ($n$ = 324; 4.5%), The Gambia ($n$ = 354; 4.9%), and Togo ($n$ = 683; 9.4%) (Table 1).

In total, 48.6% ($n$ = 3526) are living in West Africa, 31.1% ($n$ = 2255) in Central Africa, and 20.4% (1478) in Southern Africa (Table 2). Overall, 17.4% (1265/7259) of participants are living in countries where legal status of selling sex is not specified; 26.3% (1907/7259) where sex work is partially legalized, and 56.3% (4087/7259) where sex work is criminalized. Participants living in

**Table 1 Summary of data collection.**

| Region | Country | Recruitment dates | Country sample size | Study sites | Recruitment seeds | Total enrolled by site |
|---|---|---|---|---|---|---|
| West Africa | Burkina Faso | January–August 2013 | 698 | Bobo Dioulasso | 3 | 350 |
| | | | | Ouagadougou | 6 | 348 |
| | Senegal | February–November 2015 | 758 | Dakar | 9 | 502 |
| | | | | Mbour | 3 | 256 |
| | Côte d'Ivoire | March–October 2015 | 466 | Abidjan | 5 | 466 |
| | Guinea-Bissau | September 2017–January 2018 | 567 | Bafatá | 3 | 140 |
| | | | | Bissau | 8 | 323 |
| | | | | Bissorã | 3 | 45 |
| | | | | Gabu | 3 | 59 |
| | The Gambia | May 2017–May 2018 | 354 | Banjul | 9 | 354 |
| | Togo | January–June 2013 | 683 | Kara | 5 | 329 |
| | | | | Lome | 5 | 354 |
| Central Africa | Cameroon | November 2015–October 2016 | 2255 | Yaoundé | 2 | 574 |
| | | | | Douala | 1 | 457 |
| | | | | Bamenda | 1 | 341 |
| | | | | Bertoua | 1 | 304 |
| | | | | Kribi | 1 | 579 |
| Southern Africa | Lesotho | February–September 2014 | 744 | Maseru | 7 | 410 |
| | | | | Maputsoe | 12 | 334 |
| | Kingdom of eSwatini | August–October 2011 | 324 | Manzini | 14 | 324 |
| | South Africa | October 2014–April 2015 | 410 | Port Elizabeth | 9 | 410 |

**Table 2 Study sample by region, legal status of sex work, and country-level HIV epidemic.**

| Characteristics | Total (N = 7259) | |
|---|---|---|
| | **N** | **%** |
| Region | | |
| Western Africa | 3526 | 48.6 |
| Central Africa | 2255 | 31.1 |
| Southern Africa | 1478 | 20.4 |
| Legal status of sex work | | |
| Legality not specified | 1265 | 17.4 |
| Partially legalized | 1907 | 26.3 |
| Criminalized | 4087 | 56.3 |
| Country-level HIV epidemic | | |
| Generalized | 5781 | 79.6 |
| Concentrated | 1478 | 20.4 |

countries with generalized HIV epidemics represent 79.6% (5781/7259) of the study sample with 20.4% (1478/7259) in concentrated HIV epidemics.

**Demographic characteristics, HIV, disclosure, and stigmas**. Demographic characteristics, HIV risk and status, disclosure, and stigmas are summarized in Table 3. The median age is 27 years (interquartile range (IQR) 22–34), and the median years engaged in sex work is 4 (IQR 2–8). The pooled HIV prevalence is 28.6% (95% Confidence Interval (95% CI): 27.6–29.7; N = 2070/7230).

**Prevalent HIV infection and legal status of sex work**. HIV prevalence in contexts with partial legalization is 11.6% (219/1894), 19.6% (248/1265) within contexts without legal specification, and 39.4% (1603/4071) within criminalized settings (Table 4). Legal status of sex work is associated with HIV ($X^2$ p value < 0.001). When compared with settings with partial legalization, criminalized status (adjusted odds ratio [aOR]: 7.17; 95%

CI: 2.71–18.95; p value < 0.001), and sex work not being legally specified (aOR: 2.35; 95% CI: 1.06–5.21; p value = 0.036) are associated with increased odds of HIV.

Sensitivity analysis using a random sample of data from Cameroon showed similar results to the main sample (Supplementary Table 1).

**Stigmas and prevalent HIV infection**. Prevalent HIV infection is positively associated verbal harassment (aOR: 1.29; 95% CI: 1.12–1.50; p value = 0.001); blackmail (aOR: 1.39; 95% CI: 1.20–1.61; p value < 0.001); physical violence (aOR: 1.23; 95% CI: 1.02–1.49; p value = 0.029); and forced sex (aOR: 1.32; 95% CI: 1.13–1.54; p value < 0.001); and negatively associated with fear of being in public places (aOR: 0.67; 95% CI: 0.48–0.94; p value = 0.024) (Table 5).

Stigma exposure differs between participants with prior knowledge of living with HIV compared with those without prior knowledge of living with HIV or not living with HIV (Supplementary Table 2).

**HIV and stigmas by legal status of sex work**. The degree of association between stigmas and HIV varies by legal status of sex work (Mantel–Haenszel test of homogeneity (MH) p value: < 0.01) for all stigma measures assessed except denial of health services, verbal harassment, and forced sex. Specifically, in criminalized settings HIV is associated with fear of seeking health services (aOR: 95% CI: 1.01–1.53; p value: 0.041) and mistreatment in a healthcare setting (aOR: 2.15; 95% CI: 1.43–3.23; p value < 0.001), compared with partially legalized settings (Table 6).

The relationship between HIV and uniformed officers refusal to provide protection varies by legal status (MH p value < 0.01) with an increased odds in settings without legal specification (aOR: 1.64; 95% CI: 1.29–2.08; p value < 0.001) and criminalized settings (aOR: 1.38; 95% CI: 1.10–1.72; p value = 0.005) compared with partially legalized settings. Blackmail is associated with HIV in non-specified settings (aOR: 1.50; 95% CI: 1.37–1.65; p value: < 0.001) and criminalized settings (aOR: 1.35; 95% CI: 1.07–1.71; p value: 0.010) compared with partially legalized

**Table 3 Demographic characteristics, HIV risk and infection, disclosure, and stigma by legal status of sex work.**

| | Total | | | Legal status of sex work | | | | | | | X² p value |
| --- | --- | --- | --- | --- | --- | --- | --- | --- | --- | --- | --- |
| | | | | Partially legal | | Not specified | | Criminalized | | | |
| **Median Age (IQR)** | 27 (22-34) | | | 27 (23-35) | | 25 (21-30) | | 27 (23-34) | | | <0.001 |
| **Median years in sex work (IQR)** | 4 (2-8) | | | 5 (2-9) | | 4 (2-8) | | 4 (2-8) | | | |
| | n/N | % | | n/N | Column % | n/N | Column % | n/N | Column % | | |
| Age | | | | | | | | | | | <0.001 |
| 18-24 | 2689/7236 | 37.2 | | 648/1897 | 34.2 | 625/1265 | 49.4 | 1416/4074 | 34.8 | | |
| 25-30 | 2050/7236 | 28.3 | | 530/1897 | 27.9 | 342/1265 | 27.0 | 1178/4074 | 28.9 | | |
| 31+ | 2497/7236 | 34.5 | | 719/1897 | 37.9 | 298/1265 | 23.6 | 1480/4074 | 36.3 | | |
| Education level | | <0.001 | | | | | | | | | <0.001 |
| None | 1387/7229 | 19.2 | | 787/1898 | 41.5 | 338/1247 | 27.1 | 262/4084 | 6.4 | | |
| Some primary | 1625/7229 | 22.5 | | 495/1898 | 26.1 | 329/1247 | 26.4 | 801/4084 | 19.6 | | |
| Primary completed/some secondary | 3339/7229 | 46.2 | | 519/1898 | 27.3 | 495/1247 | 39.7 | 2325/4084 | 56.9 | | |
| Completed secondary or post-secondary | 878/7229 | 12.2 | | 97/1898 | 5.1 | 85/1247 | 6.8 | 696/4084 | 17.0 | | |
| Marital status | | <0.001 | | | | | | | | | <0.001 |
| Currently married | 166/7242 | 2.3 | | 48/1906 | 2.5 | 67/1264 | 5.3 | 51/4072 | 1.3 | | |
| Not currently married | 7076/7242 | 97.7 | | 1858/1906 | 97.5 | 1197/1264 | 94.7 | 4021/4072 | 98.8 | | |
| Years in sex work | | <0.001 | | | | | | | | | <0.001 |
| <5 | 3781/7001 | 54.0 | | 897/1834 | 48.9 | 665/1202 | 55.3 | 2219/3965 | 56.0 | | |
| 5 or more | 3220/7001 | 45.0 | | 937/1834 | 51.1 | 537/1202 | 44.7 | 1746/3965 | 44.0 | | |
| HIV status | | <0.001 | | | | | | | | | <0.001 |
| Living with HIV | 2070/7230 | 28.6 | | 219/1894 | 11.6 | 248/1265 | 19.6 | 1603/4071 | 39.4 | | |
| Not living with HIV | 5160/7230 | 71.4 | | 1675/1984 | 88.4 | 1017/1265 | 80.4 | 2468/4071 | 60.6 | | |
| Knowledge of living with HIV | | <0.001 | | | | | | | | | <0.001 |
| Yes | 1207/6022 | 20.0 | | 68/1410 | 4.8 | 108/720 | 15.0 | 1031/3892 | 26.5 | | |
| No | 4815/6022 | 80.0 | | 1342/1410 | 95.2 | 612/720 | 85.0 | 2861/3892 | 73.5 | | |
| Disclosure of sex work to family | | <0.001 | | | | | | | | | <0.001 |
| Yes | 1627/7250 | 22.5 | | 409/1902 | 21.5 | 188/1265 | 14.9 | 1030/4083 | 25.2 | | |
| No | 5623/7250 | 77.6 | | 1493/1902 | 78.5 | 1077/1265 | 85.1 | 3053/4083 | 74.8 | | |
| Disclosure of sex work to healthcare provider | | <0.001 | | | | | | | | | <0.001 |
| Yes | 1401/3636 | 22.0 | | 710/1827 | 38.9 | 255/1262 | 20.2 | 436/3274 | 13.3 | | |
| No | 4962/6363 | 78.0 | | 1117/1827 | 61.1 | 1007/1262 | 79.8 | 2838/3274 | 86.7 | | |
| Stigma | | | | | | | | | | | |
| Family exclusion | 914/7176 | 12.7 | | 204/1897 | 10.8 | 186/1208 | 15.4 | 524/4071 | 12.9 | | 0.001 |
| Family gossip | 1450/7203 | 20.1 | | 452/1887 | 24.0 | 281/1253 | 22.4 | 717/4063 | 17.7 | | <0.001 |
| Friend rejection | 962/7130 | 13.5 | | 216/1792 | 12.1 | 173/1260 | 13.7 | 573/4078 | 14.1 | | 0.109 |
| Afraid of seeking health services | 968/7248 | 13.4 | | 337/1902 | 17.7 | 171/1262 | 13.6 | 460/4084 | 11.3 | | <0.001 |
| Avoided seeking health services | 673/6572 | 10.2 | | 322/1901 | 16.9 | 125/1264 | 9.9 | 226/3407 | 6.6 | | <0.001 |
| Mistreated in health center | 190/7170 | 2.7 | | 67/1821 | 3.7 | 15/1264 | 1.2 | 108/4085 | 2.6 | | <0.001 |
| Health care provider gossip | 317/7169 | 4.4 | | 115/1823 | 6.3 | 35/1264 | 2.8 | 167/4082 | 4.1 | | <0.001 |
| Denied health services | 94/7244 | 1.3 | | 33/1894 | 1.7 | 13/1265 | 1.0 | 48/4085 | 1.2 | | 0.127 |
| Police refused protection | 1036/7096 | 14.6 | | 225/1754 | 12.8 | 139/1261 | 11.0 | 672/4081 | 16.5 | | <0.001 |
| Scared in public places | 982/6275 | 15.7 | | 479/1904 | 25.2 | 158/696 | 22.7 | 345/3675 | 9.4 | | <0.001 |
| Verbally harassed | 3140/6576 | 47.8 | | 818/1904 | 43.0 | 494/1265 | 39.1 | 1828/3407 | 53.7 | | <0.001 |
| Blackmailed | 2198/7249 | 30.3 | | 516/1905 | 27.1 | 256/1262 | 20.3 | 1426/4082 | 34.9 | | <0.001 |
| Physical violence* | 2359/7240 | 32.6 | | 607/1905 | 31.9 | 459/1256 | 36.5 | 1293/4079 | 31.7 | | 0.004 |
| Forced to have sex* | 2207/7234 | 30.5 | | 597/1905 | 31.3 | 343/1263 | 27.2 | 1267/4066 | 31.2 | | 0.017 |

*Not specified as attributable to sex work.

**Table 4 HIV infection and country-level legal status.**

| Legal status of sex work | Living with HIV | | | | | | | |
|---|---|---|---|---|---|---|---|---|
| | n/N | % | OR | P value | 95% CI | aOR* | P value | 95% CI |
| Partially legalized | 219/1894 | 11.6 | Ref. | Ref. | Ref. | Ref. | Ref. | Ref. |
| Selling not specified | 248/1265 | 19.6 | 1.87 | 0.181 | 0.78, 4.65 | 2.35 | 0.036 | 1.06, 5.21 |
| Criminalized | 1603/4071 | 39.4 | 4.97 | 0.001 | 1.98, 12.44 | 7.17 | < 0.001 | 2.71, 18.95 |

*Adjusted for age, education level, marital status, years in sex work, clustered by site and by country.

**Table 5 Pooled relationship between stigma and prevalent HIV infection among female sex workers.**

| Stigmas | | Living with HIV | | | | | |
|---|---|---|---|---|---|---|---|
| | | OR | 95% CI | P value | aOR** | 95% CI | P value |
| Perceived | Family exclusion | 1.68 | 1.46, 1.94 | <0.001 | 1.05 | 0.84, 1.31 | 0.686 |
| Perceived | Family gossip | 1.33 | 1.17, 1.50 | <0.001 | 0.95 | 0.78,1.16 | 0.637 |
| Perceived | Friend rejection | 1.88 | 1.64, 2.17 | <0.001 | 1.28 | 0.98, 1.67 | 0.068 |
| Anticipated | Afraid of seeking health services | 1.22 | 1.05, 1.41 | 0.008 | 0.97 | 0.72, 1.30 | 0.824 |
| Anticipated | Avoided seeking health services | 0.75 | 0.62, 0.91 | 0.003 | 0.79 | 0.48, 1.30 | 0.358 |
| Perceived | Mistreated in health center | 1.82 | 1.35, 2.43 | <0.001 | 1.09 | 0.44, 2.72 | 0.849 |
| Enacted | Health care provider gossip | 1.17 | 0.92, 1.49 | 0.197 | 0.95 | 0.49, 1.83 | 0.832 |
| Enacted | Denied health services | 1.48 | 0.97, 2.26 | 0.066 | 1.10 | 0.56, 2.17 | 0.792 |
| Perceived | Police refused protection | 2.14 | 1.87, 2.45 | <0.001 | 1.26 | 0.97, 1.64 | 0.141 |
| Perceived | Scared in public places | 0.96 | 0.83, 1.13 | 0.673 | 0.67 | 0.48, 0.94 | 0.024 |
| Enacted | Verbally harassed | 1.32 | 1.18, 1.47 | <0.001 | 1.29 | 1.12, 1.50 | 0.001 |
| Enacted | Blackmailed | 1.19 | 1.07, 1.33 | 0.002 | 1.39 | 1.20, 1.61 | <0.001 |
| Enacted | Physical violence* | 1.58 | 1.42, 1.76 | <0.001 | 1.23 | 1.02, 1.49 | 0.029 |
| Enacted | Forced to have sex* | 1.29 | 1.16, 1.44 | <0.001 | 1.32 | 1.13, 1.54 | <0.001 |

Each stigma indicator assessed through a separate model due to collinearity between stigma items.
*Not specified as attributable to sex work.
**Adjusted for age, education level, marital status, years in sex work, country-level epidemic, and clustered by site and by country. Adjusted for disclosure of sex work to family of healthcare provider when conceptual relevant (social stigma; healthcare-related stigma).

**Table 6 Relationship between stigma and HIV by legal status of sex work.**

| Stigmas | HIV infection | | | | | | | |
|---|---|---|---|---|---|---|---|---|
| | Test for interaction | Partially legalized | Selling not specified | | | Criminalized | | |
| | MH test of homogeneity p value^ | Reference category | aOR** | 95% CI | P value | aOR** | 95% CI | P value |
| Family exclusion | 0.0002 | — | 0.90 | 0.53, 1.52 | 0.699 | 1.13 | 0.85, 1.50 | 0.417 |
| Family gossip | <0.0001 | — | 0.76 | 0.54, 1.06 | 0.102 | 1.14 | 0.86, 1.51 | 0.353 |
| Friend rejection | 0.0062 | — | 1.29 | 0.89, 1.87 | 0.185 | 1.27 | 0.93, 1.74 | 0.126 |
| Afraid of seeking health services | 0.0001 | — | 1.28 | 0.81, 2.03 | 0.285 | 1.24 | 1.01, 1.53 | 0.041 |
| Avoided seeking health services | 0.0074 | — | 1.44 | 0.90, 2.29 | 0.130 | 1.18 | 0.81, 1.72 | 0.376 |
| Mistreated in health center | 0.0016 | — | 1.54 | 0.51, 4.63 | 0.446 | 2.15 | 1.43, 3.23 | <0.001 |
| Health care provider gossip | 0.0039 | — | 0.86 | 0.25, 2.91 | 0.804 | 1.46 | 0.78, 2.72 | 0.227 |
| Denied health services | 0.1511 | — | — | — | — | — | — | — |
| Police refused protection | 0.0005 | — | 1.64 | 1.29, 2.08 | <0.001 | 1.38 | 1.10, 1.72 | 0.005 |
| Scared in public places | <0.0001 | — | 0.87 | 0.62, 1.21 | 0.400 | 0.91 | 0.65, 1.25 | 0.543 |
| Verbally harassed | 0.1687 | — | — | — | — | — | — | — |
| Blackmailed | 0.0025 | — | 1.50 | 1.37, 1.65 | <0.001 | 1.35 | 1.07, 1.71 | 0.010 |
| Physical violence* | <0.0001 | — | 0.79 | 0.62, 1.01 | 0.070 | 1.34 | 1.11, 1.62 | 0.002 |
| Forced to have sex* | 0.0275 | — | — | — | — | — | — | — |

Each stigma indicator assessed through a separate model.
Mantel–Haenszel test of homogeneity was used to assess effect measure modification between types of stigma and legal status of sex work across each stigma exposure model assessing the association with HIV status. Those values for which a statistical interaction was observed were assessed through stratified multivariable adjusted models assessing the impact of stigma exposures on HIV status.
^<0.01 significance level for test for homogeneity.
*Not specified as attributable to sex work.
**Adjusted for age, education level, marital status, years in sex work, country-level epidemic, and clustered by site and by country. Adjusted for disclosure of sex work to family of healthcare provider when conceptual relevant (social stigma, and healthcare-related stigma).

settings. HIV is associated with physical violence in criminalized settings versus partially legalized settings (aOR: 1.34; 95% CI: 1.11–1.62; $p$ value = 0.002).

## Discussion

Punitive and non-protective sex work laws are associated with prevalent HIV infection among female sex workers in countries across sub-Saharan Africa. The prevalence of stigma is high among female sex workers and consistently associated with prevalent HIV infection, highlighting the importance of structural determinants alongside more proximal individual-level characteristics. The degree of the relationship between stigmas and HIV varies by legal status of sex work, suggesting that stigmas and legal status of sex work may operate jointly in increasing individual HIV burden. This study further demonstrates the persistence of certain types of stigmas across differing legal contexts and suggests that the potential impact of stigmas on HIV risk and ultimately burden may be greatest in punitive and non-protective settings. Finally, these results suggest the complexity of HIV risks among sex workers across sub-Saharan Africa transcending individual-level sexual practices, highlighting the need to continue to measure and address stigmas to inform a more effective and efficient HIV response.

The magnitude of the relationship between the legal status of sex work and individual HIV infection is highest among individuals in fully criminalized settings, followed by settings where the legal status of selling sex is not specified. These results are consistent with prior findings from ecological studies[19] and highlight how laws serve as a structural determinant that contribute to individual-level health outcomes. Findings suggest that written laws, independent of enforcement practices, influence HIV outcomes and that explicit legality serves in protecting sex workers. Moreover, these results suggest that punitive and non-protective laws may contribute to an environment that perpetuates HIV risks among sex workers. These findings are consistent with earlier mathematical models that demonstrated that across generalized and concentrated HIV epidemics, decriminalization of sex work could have the largest effect on the course of country-level epidemics, averting one-third to almost one-half of incident HIV infections over the next decade[10]. This reduction would be through combined effects on violence, harassment by uniformed officers, and safer work environments collectively mediating HIV transmission pathways[10]. Despite these consistent results, the number of countries decriminalizing sex work has not increased over the last 5 years[2,11].

The relationship between stigmas and HIV varies across different legal contexts of sex work, suggesting that stigmas and sex work laws interact in increasing HIV risks and ultimately burden. Sex workers living in settings with criminalized and non-specified laws generally show a stronger relationship between stigmas and HIV compared with partially legalized settings. Existing evidence suggests that sex workers living in punitive and non-protective settings may experience greater burdens of stigmas than women living in partially legalized settings[37]. However, in this study, women reporting any lifetime history of stigma is not clearly or consistently higher in criminalized or non-protective settings compared with partially legalized settings, highlighting that sex workers across legal environments experience stigmas. Although sex workers may still experience a greater frequency of stigmas over the course of their lifetime in punitive and non-protective settings, the periodicity of stigma experiences among women is not measured in this study. Given the near universality of stigmas affecting sex workers, the mechanisms associated with increased HIV burden may act by amplifying the barriers to safety, as well as efficacious health services. Specifically, sex workers in punitive and non-protective environments may be more susceptible to the harms related to stigmas affecting overall safety in society and in access to HIV prevention and treatment services. Furthermore, sex workers in partially legalized or more protective environments have been shown to have higher levels of social capital, resiliency, and options for support that can mitigate the impact of stigmas on HIV risks[38]. Ultimately, the mechanisms underpinning the synergies of stigmas and sex work laws in the burden of HIV among sex workers likely vary by the specific type of stigma. The consistency in the findings of the interaction between laws and stigma in especially punitive legal settings reinforce the importance of HIV prevention and treatment intervention strategies tailored for sex workers that consider the legal context during implementation.

Uptake of HIV testing, prevention, and treatment services remains low among sex workers across sub-Saharan Africa and globally, in part due to healthcare-related stigmas[2,10,39]. In these analysis, higher HIV burden among sex workers was associated with anticipated and perceived stigmas relating to seeking care in criminalized settings. Harmful effects of stigma are reinforced by the experience of intersecting stigmas among sex workers living with HIV attributable to both sex work and HIV status, as participants who reported to be aware of living with HIV prior to enrollment report higher levels of sex work-related stigmas. The combined or compounded effect of multiple stigmas may further influence uptake of services and health outcomes[40–42]. Leveraging innovative approaches to provide services outside of health facilities while working to mitigate observed individual and intersecting stigmas may facilitate improved service coverage. In this context, decentralized services have been able to serve sex workers who were not accessing traditional services and therefore may provide an avenue to increase coverage and access[43–45]. To date, few studies have evaluated stigma mitigation approaches for people living with HIV and even fewer have aimed to study stigma reduction for sex workers in healthcare settings across sub-Saharan Africa[24]. These results suggest the importance of protective structures within healthcare systems, such as the enforcement of anti-discriminatory policies and accountability mechanisms to ensure culturally and clinically competent services for all.

Violence affecting sex workers is prevalent across legal contexts and is associated with HIV among sex workers in this analysis, consistent with findings from other settings[46]. Violence has been associated with HIV risks, such as inconsistent condom use, difficultly in condom negotiation, recent condom failure, client condom refusal, and high client volume[47–49]. In this analysis, the relationship between physical violence and HIV varies by legal context, with an increased association in criminalized settings. Increased legal restrictions on sex work has been shown to move activities to more hidden settings to avoid detection by uniformed officers, alongside increased vulnerability to violence and HIV-risk behaviors such as unprotected sex[50]. Even when enforcement efforts prioritize clients or third parties, violence affecting sex workers persists[51]. In contrast, the degree of the relationship between sexual violence and HIV does not vary across legal contexts. Aligning with findings from other studies, this suggests that partial legalization, such as the removal of only some aspects of criminal laws and regulation of sex workers is necessary, but not sufficient for reducing sexual violence as a risk factor for HIV[52]. This is consistent with previous modeling and empirical work, suggesting that only through full decriminalization, such as full removal of laws targeting sex industry; access to safer work environments; and prevention of violence and harassment by police could law reform as a structural determinant avert violence and HIV infections[10,51]. Finally, empirical research pre and post law reform has shown that in settings where clients and third

parties are criminalized but not sex workers, rates of both sexual violence was unchanged from full criminalization[53].

Sex workers reporting lack of protection from uniformed officers is prevalent in this analysis, and likely true in much of the world[54]. The lack of protection explains the persistent violence and blackmail observed among sex workers across legal contexts, likely due to impunity of offenders. There have been limited recent efforts among countries or regions to end impunity for crimes and abuses against sex workers[2]. In this study, the relationship between blackmail and HIV is highest in non-protective settings, followed by criminalized settings. In other settings, repressive police practice has been associated with violence, as well as HIV and sexually transmitted infections[18]. Perceived stigmas related to policing practices is prevalent and associated with HIV infection in this analysis, suggesting that legal protections as well as training and accountability of law enforcement may support improved HIV outcomes. Here, there is no assessment of enacted stigmas by uniformed officers specifically, however, this has been observed in other settings[55]. For instance, qualitative assessments have reported abusive practices by uniformed officers against sex workers, including blackmail, arbitrary arrest, and violence[56,57]. Women have also reported that sex or money are used as compensation for release after arrests[56,57]. Among sex workers in Côte d'Ivoire, Burkina Faso, and Togo who had experienced physical violence and forced sex, a large proportion reported perpetration by a uniformed officer[55,58]. In this study, fear of being in public places is negatively associated with HIV prevalence overall, prior to stratification across legal contexts. In part, the lower HIV risk may emerge from protective behaviors such as avoiding street-based sex work, which has been associated with increased violence, extortion by uniformed officers, and increased HIV-related risk behaviors[18,59]. Combined structural interventions with uniformed officers involving advocacy with senior uniformed officers, and crisis response mechanisms have been shown to reduce uniformed officers arrests and violence, and create a safer work environment for sex workers[60,61].

Several limitations in this study should be considered. Although data are clustered by site and country to account for the non-independent nature of observations within each site and within each country, individual country and site differences may be lost in the aggregation of data. Legal status categories were determined based on country-specific legislation where available, but not necessarily the enforcement practices and justice systems. Although we are not able to assess causality through cross-sectional data and cannot account for the relationship between HIV prevalence and stigmas over time, laws were established prior to HIV introduction within countries. At a minimum, this limits the possibility that laws criminalizing sex work were a result of or influenced by the HIV epidemic. It is possible that unmeasured confounders preceding both sex work laws and country-level epidemics may exist and feed independently into both, thus resulting in uncontrolled confounding. There may also be unmeasured confounders that are associated with sex work laws and/or stigmas, as well as causally associated with HIV, but not on the casual pathway between these exposures and outcomes. None of the countries included in this analysis meet the criteria for decriminalized legal status, and therefore this legal context could not be evaluated. Data were collected over a period of 7 years, which should be considered in the interpretation of the results. Enforcement practices, program funding, and other external measures over time may have influenced stigmas, HIV status, or HIV risk. Female sex workers living with HIV may experience intersectional or compounded stigmas due to HIV status and engagement in sex work, and therefore there is a need to further evaluate these intersectional stigmas.

In the context of a slowing HIV pandemic, epidemics among sex workers in most settings across sub-Saharan Africa are stable or growing. Although others are benefiting from improved prevention and treatment interventions, these data highlight that both sex work laws and stigmas prevent the effective provision and uptake of interventions for sex workers across sub-Saharan Africa. Moreover, the unmet HIV treatment needs among sex workers results in onward HIV transmissions that are relevant even in the most generalized HIV epidemics. The data presented here collectively demonstrate the importance of punitive and non-protective laws in driving HIV risks among sex workers. Furthermore, stigmas and sex work laws appear to operate synergistically in increasing HIV burden, with stigmas having a greater impact on HIV risk in punitive and non-protective settings. In 2020, there will be more than three times the number of infections compared with the stated goal of 500,000 new HIV infections, highlighting the need to do things differently if there is to be a chance of achieving zero new infections by 2030. Thus, whether the path forward is driven by human rights or public health principles, achieving zero new HIV infections in the foreseeable future can only be realized if we meaningfully address the structural determinants that contextualize individual HIV risks among sex workers across sub-Saharan Africa.

## Methods

**Data collection and participants**. Primary data collection was conducted through 10 country-specific studies led by one investigative team. Respondent driven sampling (RDS) was used in each of the 10 country-specific studies between 2011 and 2018. All country-specific studies were cross-sectional. Data were collected across 21 sites in 10 countries: Burkina Faso (January–August 2013); Cameroon (November 2015–October 2016); Côte d'Ivoire (March 2015–February 2016); The Gambia (May 2017–May 2018); Guinea-Bissau (September–November 2017); Lesotho (February–September 2014); Senegal (February–November 2015); eSwatini (August–October 2011); South Africa (October 2014–April 2015); and Togo (January–June 2013).

RDS, a peer-recruitment method designed to sample marginalized populations, was administered independently across the country-specific sites to recruit female sex workers. Recruitment chains were initiated by seeds in each site, who were individuals selected in collaboration with local community-based organizations to represent heterogeneity in demographic characteristics and geographic representation. Initial seeds were provided with three coupons to recruit peers into the study. Women recruited by seeds and enrolled in the study were provided with three coupons for continued recruitment of peers. This process was repeated until reaching the target sample size of each country. Sample size calculations for the initial data collection were powered to estimate HIV prevalence at each site. The number of recruitment seeds by study site are provided in Table 1.

Participants were eligible if they self-reported female sex assigned at birth; were 18 years or older; attributed more than half of their income in the past 12 months to selling sex; and were capable of providing informed consent. Country-specific eligibility criteria included city or area of residence. All participants provided verbal or written informed consent. The study complied with all relevant ethical regulations for work with human participants. Country-specific data collection were reviewed and approved by Johns Hopkins School of Public Health Institutional Review Board and/or an ethical review board and related bodies in the country of data collection. Country-specific ethic committees include: Health Research Ethics Committee of Burkina Faso, National Ethics Committee of Cameroon, the Health Research Ethics Committee of Côte d'Ivoire, National Research Ethics Committee of Guinea Bissau, the Lesotho National Health Research Ethics Committee, the Senegalese National Health Research Ethics, Institutional Review Boards of the Human Sciences Research Council in South Africa, the Swaziland Scientific Ethics Committee, Scientific Coordination Committee in the Gambia, the Ethical Committee of Togo.

Interviewer-administered questionnaires were conducted, and socio-behavioral measures were self-reported. All interviews were conducted in a private location with trained study staff. Biological testing for HIV, including pre- and post-test counseling, was conducted consistent with country-specific national guidelines. Participants with a reactive test result were referred to care. Pre-test counseling and biological testing were conducted prior to administering the socio-behavioral questionnaires. Post-test counseling and HIV test results were reviewed with participants after completion of the socio-behavioral questionnaire.

**Measures**. Individual-level data from socio-behavioral questionnaires and biological testing are used for this analysis. Stigma measures are described in Supplementary Table 3. Consistent stigma metrics are used across countries that included items on ever experiencing stigma relating to healthcare, among family or friends,

and the general community. Stigma measures were asked as attributable to engagement in sex work, except for measures of physical and sexual violence. Stigma measures were informed by a systematic review of stigma metrics and were validated with data from Togo and Burkina Faso[26,62,63].

Countries are categorized into three regions: West Africa (Burkina Faso, Cote d'Ivoire, The Gambia, Guinea-Bissau, Senegal, and Togo); Central Africa (Cameroon); and Southern Africa (Lesotho, Kingdom of eSwatini, South Africa). Country-level HIV epidemic status is defined for each country as either generalized or concentrated. These categories leveraged the traditional UNAIDS and WHO definitions. Thus, a concentrated HIV epidemic includes countries in which HIV prevalence is consistently over 5% in at least one defined subpopulation, but < 1% among reproductive aged women; a generalized HIV epidemic has an HIV prevalence consistently exceeding 1% in adult women. UNAIDS estimates were used to categorize the country-level epidemics[64].

Legal status of sex work for countries in this study is defined and categorized based on the legal approach: not specified, partially legalized, or criminalized (Supplementary Table 4). Not specified included countries in which there is not an explicit law legalizing or criminalizing the selling of sex. Partial legalization is defined as countries that have legalized an aspect or a mechanism of sex work under specific circumstances, including legal to sell or legal to solicit, whereas leaving other aspects criminalized. In some cases, legalization of sex work is regulated alongside a registration system for sex workers. Criminalized, is defined as illegal to sell sex, solicit sex, and organize commercial sex under any circumstance and stipulated punishment under the law. None of the countries in these analyses was considered decriminalized. This categorization was determined by leveraging existing legal frameworks for sex work[18,65,66]. The legal status of sex work for this analysis is defined by the written law and not based on enforcement practices.

**Statistical analyses.** Statistical models were guided by the Structural HIV Determinants Framework for Sex Work and the Logic Model of Public Health Law Research[10,67].

Data are pooled across countries and analyzed as crude data; RDS-adjusted weighting is not applied across countries as women do not represent a single network of female sex workers, violating a key assumption of RDS[68]. Models are clustered by country and by site and represent valid sample estimates, but may differ from population-level estimates given lack of RDS-adjustment[69].

Proportions of demographic characteristics, HIV risk and status, disclosure, and stigmas are described using crude estimates. Person's Chi-squared is used to assess the relationship between demographic characteristics and legal status of sex work.

Legal status of sex work and HIV are assessed through simple and multivariable logistic regressions. Multivariable logistic regression models adjusted for age, education level, marital status, and years in sex work and account for clustering within sites and within countries. Although the country-level epidemic (concentrated vs. generalized) is associated with HIV prevalence, it is not considered a confounder in our conceptual model, but rather a mediator between sex work law and HIV prevalence, as sex work laws in each country preceded the introduction of HIV within the countries.

Logistic regression models are used to assess associations between various stigma exposures and the outcome of HIV status. Stigma exposure models were run separately due to collinearity between stigma items. Multivariable logistic regression models adjusting for country-level epidemic, age, education, marital status, and years in sex work, and respective disclosure variables when conceptually relevant (disclosure of sex work to family; disclosure of sex work to healthcare provider) were run for each stigma exposure. All models account for clustering by site and by country.

To explore joint mechanisms through which the relationship between stigma indicators and HIV status may be modified by legal status, the relationship between stigma indicators and HIV status is stratified by country-level legal status of sex work. The MH is used to assess differences between stigma and HIV across different legal statuses, using a significance level of $p < 0.01$[70]. Values for which a difference was observed were assessed through stratified multivariable adjusted models assessing stigma exposures on HIV status. Effect measure modification between stigma and HIV by legal status of sex work was assessed using an interaction term of stigma and legal status in logistic regressions with HIV modeled as the outcome. Effect measure models adjust for country-level epidemic, age, education, marital status, years in sex work, and respective disclosure variable when conceptually relevant; models account for clustering within sites and within countries.

Due to the large sample size from Cameroon, sensitivity analyses using a random sample of data from Cameroon ($n = 700$) were conducted.

All analyses were conducted in Stata v.15.1. (College Station, Texas, United States).

**Reporting summary.** Further information on research design is available in the Nature Research Reporting Summary linked to this article.

## Data availability
The data that support the findings of this study are available from the corresponding author upon reasonable request.

## Code availability
The custom code in these analyses are available from the corresponding author upon reasonable request.

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

## Acknowledgements

We express our sincere appreciation to the participants of this study. In addition, we acknowledge the crucial role of the community groups that make great personal and professional sacrifices to serve the unmet health and advocacy needs of those most marginalized in the HIV response. We would also like to thank the data collection and study coordination teams across the different countries. The work was funded through USAID, PEPFAR, Global Fund to Fight AIDS, Tuberculosis and Malaria, and NIH. Work in Togo and Burkina Faso was supported by Project SEARCH, which was funded by the US Agency for International Development (USAID) under Contract GHH-I-00-07-00032-00 and by the President's Emergency Plan for AIDS Relief (PEPFAR). Cameroon study was supported through the CHAMP project, which was led by CARE and funded by PEPFAR through USAID. Work in Côte d'Ivoire was funded by the Global Fund to Fight AIDS, Tuberculosis and Malaria through the Government of Côte d'Ivoire National AIDS Control Program (PNPEC) contract to Enda Santé, and subcontracted for technical assistance to Johns Hopkins University. Work in Guinea Bissau and The Gambia was funded through Global Fund to Fight AIDS, Tuberculosis and Malaria. Work in Senegal was funded through HIV Prevention 2.0 (HP2): Achieving an AIDS-Free Generation in Senegal and supported by the USAID under Cooperative Agreement No. AID-OAA-A-13-00089. Work in Lesotho was funded by USAID (AID-674-A-00-00001) and implemented by Population Services International/Lesotho (PSI). Work in eSwatini was funded by PEPFAR through the USAID Swaziland (GHH-I-00-07-00032-00). Work in South Africa was funded in part by a grant provided by the MAC AIDS Fund (grant No.GR-000001400). C.E.L.'s effort was supported by the Johns Hopkins HIV Epidemiology and Prevention Sciences Training Program (5T32AI102623-08). S.B.'s effort was supported by the National Institute Of Mental Health of the National Institutes of Health under award number R01MH110358; and the National Institute Of Nursing Research of the National Institutes Of Health under award number R01NR016650. Publication was

supported by The Foundation for AIDS Research (amfAR); the National Institute Of Mental Health of the National Institutes of Health under award number R01MH110358; the National Institute Of Nursing Research of the National Institutes Of Health under award number R01NR016650; The Linkages across the Continuum of HIV Services for Key Populations Affected by HIV (LINKAGES) project funded by PEPFAR and USAID and led by FHI360; and the CHAMP project. Finally, this publication was made possible by the Johns Hopkins University Center for AIDS Research, an NIH funded program (P30AI094189). The funders had no role in study design, data collection and analysis, decision to publish, or preparation of the manuscript.

## Author contributions

C.E.L. and S.B. collaborated on the conceptualization of the study. C.E.L., S.B., S.S., S.M. collaborated in analytic plan, analyses, and interpretation. D.D., T.M., S.K., A.S., A.K., Z.M., U.T., N.P.M., B.C., F.D., M.A.D., S.B. collaborated on study design, implementation, and investigation. C.E.L. and S.B. led initial drafting of the manuscript with S.S., S.M., K.S., D.D., T.M., S.K., A.S., A.K., Z.M., U.T., N.P.M., B.C., F.D., M.A.D. contributing to specific sections and review and revisions.

## Competing interests

The authors declare no competing interests.
