## [Peer Review File · Nature Communications]

Reviewers' Comments:

Reviewer #1:

Remarks to the Author:

General Comments:

The topic of this paper is timely and important, examining structural factors and HIV risk among female sex worker through seeking to understand the relationship between the legal environment regarding sex work, stigma, and HIV prevalence in sub-Saharan Africa. It draws on a unique and valuable cross-sectional survey data set with female sex workers across ten countries and leverages country-to-country variation in sex work laws to examine the relationship between HIV prevalence, stigma and criminalization of sex workers. Its major strength is that it consists of a large sample of a difficult-to-reach population---female sex workers across multiple countries where this is likely the only data of its kind. Its major weakness is the high risk of confounding across country contexts, most of which is not controlled for in the study.

Specific comments:

Abstract

1. The abstract leads the reader to believe that there will be data presented on intersecting stigmas in the body of the text. To my understanding of intersectional stigma, none is presented in this paper, so I suggest removing the term from the abstract.
2. As this article is about female sex workers only, it would be good to qualify that in the abstract and throughout the text. (I realize it is in the title, but additional qualification throughout would help the reader)

Introduction

3. I find the use of the word potentiate, both in the title and throughout the text odd and I think it may be difficult for most readers to grasp/understand. In looking up its meaning (I didn't know what this word meant), it seems to be used almost exclusively in the content of discussing increasing potency of drugs. Perhaps consider using a different word, like increasing, intensifying etc. to make the point of the paper clearer to the average reader, both in the title and in the text.

Methods

4. Were respondents asked about their HIV status first, before they were tested for HIV? Was pre and post-test counseling done before or after the questionnaire/stigma data collection? If respondents self-reported as HIV positive, was length of knowing HIV status collected? I ask because I wonder if sex workers who have been living with HIV for a while, and so may be experiencing intersectional stigma related to both HIV status and sex work, may be reporting higher levels of sex work stigma than those who are HIV negative and/or do not know their status. (which could imply intersectional stigma?) This might be worth discussing in the discussion section. What are the potential implications, particularly with respect to the relationship found between HIV status and stigma?
5. The sampling approach requires more detail. How was RDS conducted? How many seeds in how many locations? How long were the referral chains? How were the interviews conducted, etc?
6. While mentioned that this data was collected using RDS, it is unclear if it is being analyzed as RDS. It seems, that it's being analyzed like a simple random sample (with some analyses being clustered by country). RDS sampling typically requires the application of RDS weights and some approach to account for observations' non-independence in referral chains. If the analysis doesn't incorporate weights accounting for clustering by referral chain (or another method of addressing standard errors – there's a substantial debate in the field about how best to account for observations' non-independence), it should be explained.
7. Were the data re-weighted when pooled? For example, Cameroon represents about 1/3 of the data

and more than half of the data from fully criminalized settings. Is the Cameroon sample proportional to the estimated size of the population of sex workers in each setting? Are the results sensitive to how much weight is given to the observations from each location?

8. The association between each indicator of stigma and HIV is separately modeled rather than combined because of collinearity between the items. This indicates that they are potentially capturing the same underlying construct and I would also expect that many load onto the same factor, for example, family exclusion and family gossip to be indicators of the same underlying construct. Was any factor analysis done and if so, what is the rationale for individual items versus scaling?

9. Curious about the background demographic variables that were collected. It seems only two were collected—age and education? Or were others collected and not included in the paper because they were non-significant? What was the rationale for including just these two? Was length in sex work collected? It seems that would potentially be an important variable with respect to the experience of stigma—since stigma items are asked as ever happened questions? Marital status seems like potentially another important variable, along with disclosure of sex work status.

Results

10. The associations between the legal status of sex work and HIV seem likely to be highly confounded. The same is true for associations between stigma and HIV stratified by legal status. Places where sex work is criminalized are mainly southern and central Africa; places where it is partially legalized or not specified are mainly west Africa. Without more theory and stronger control variables, one cannot know whether the observed associations are spurious and simply reflect regional differences in HIV risk, whether greater HIV prevalence led to more criminalization, etc. I am not certain what the best way to account for underlying HIV risk is – perhaps a variable for local prevalence level?

11. The customary way to assess relationships between legal status and health outcomes is to use a difference-in-differences/controlled before-after approach or, if not possible, before-after with careful statistical controls. Comparing cross-sectional data from different legal contexts is fraught because there is a high risk of confounding and reverse causation. At minimum, it requires discussion of a strong theoretical framework operationalized into control variables. (See, for example, Wagenaar's book and other materials at <http://publichealthlawresearch.org/theory-methods>.)

12. Please cross-check all the figures in the text with the tables—there are several that are different in the text compared to the tables. For example, reported verbal harassment 31.2% (2200/7252) in the text and 30.34% in table 3. There are others as well.

13. Given the detail is all in the tables, would suggest not repeating every single figure/result in the text, but instead summarizing or highlighting key trends or differences from the tables.

14. To help the reader, I would suggest putting the relevant table(s) for each section next to the header for each section, e.g. HIV and stigma (table 3 and 4); The results section is quite dense and this will help the reader follow along more easily, to have the referent table listed up at the start of each section

15. Legal status of sex work section (starts at line 160)—which is table 5---also includes association of legal status with stigma (table 6)—essentially the last line in this section only. Suggest modifying the header to say Legal status of sex work (Table 5) and stigma (table 6)

16. Line 168/169. Stigma was associated with legal status of sex work except for family gossip and denial of healthcare (table 6). Table 6 indicates that family gossip was significant, but friend rejection was not. Which is correct—what is in the text or what is in the table?

17. HIV and stigma by legal status of sex work (Table 7). In presenting the results of the regression analysis, it will be important to note that in some cases the odds ratios are below 1—indicating that that the relationship between the stigma item and HIV infection is significantly lower for the selling not specified setting versus the comparison legal setting---partially legalized. (family gossip, scared in public places and physical violence). Based on the rest of the text of the paper, I'm assuming this is an unexpected finding (that one would expect the odds to be higher in the selling not specified than

partially legalized) or maybe it is not (in which case where the odds are higher would need discussion). Some discussion of the fact that these relationships change for different stigma items and the partially legalized vs selling not specified in the discussion section is warranted.

Discussion

18. I'm not sure it's customary to call this kind of study a meta-analysis, since so far as I can tell the data haven't been published previously. Rather, it's really a multi-site survey, unless I'm missing something. Consider removing the term meta-analysis throughout the discussion section.

19. See comment #4 above.

20. See comment #17 above.

Limitations

21. The studies were conducted over a period of about seven years (which is reasonable considering the difficulty achieving large sample sizes in hard-to-sample populations). However, it would be worth discussing in the limitations section whether this poses any risk of confounding or bias. This would be particularly problematic if there are any systematic relationships expected between calendar time and legal status, stigma, or HIV risk.

Reviewer #2:

Remarks to the Author:

The issue that the authors address, the relationship of stigma and sex work laws with HIV risks among female sex workers, is timely and socially important. The data, collected by the research team across 10 sub-Saharan countries, is unique and valuable. The examination of individual-level data stratified by country-level legal status of sex work represents a sophisticated analytical approach. The results are generally strong and consistent, and supportive of expectations. These aspects of the work make it a potential candidate for publication.

However, despite my generally favorable orientation to this work because of its practical implications, I do have a number of reservations about conceptual aspects of the research. One reservation that I have concerns the magnitude of new conceptual contribution of the research. What makes this research unique is not so much the findings of a relationship between stigma and HIV risk – a number of studies of various groups already document such a relationship – but the particular focus of the current work on female sex workers and differences related to the legal status of sex work. As the authors note, a very small percentage of studies in the literature concern stigma associated with sex work. However, as the authors acknowledge, there is other work also linking laws about sex work and incidence of HIV: "Mathematical modelling suggests that across generalized and concentrated HIV epidemics, decriminalization of sex work could have the largest effect on the course of the HIV epidemic, averting one third to almost one half of incident HIV infections over the next decade." Nevertheless, I do think that the empirical evidence presented by the authors may be sufficiently unique for publication – if the authors make the nature of new contribution more conceptually compelling.

I do not believe that the current manuscript makes a strong enough conceptual case, though. My concerns about this facet of the manuscript involve both the introductory framing of and interpretation of the work. The current introduction is limited in explaining how and why stigma increases HIV risk. The authors state that stigma "results in a separation from the social norms and leads to adverse experiences and a loss of social status." While stigma is a devaluation of a person with a marked

identity, people cope with stigma in a wide range of ways – not all of which would lead to greater HIV risk. Greater stigma, for example, could lead people to different work, which might lower HIV risk. Women who responded this way are not in the current sample. In addition, the authors make a distinction between enacted and perceived stigma in their measures. This is a common and important distinction. However that distinction is only noted in passing, not really considered analytically, in the data analysis and interpretation.

Another key point that I believe needs to be developed further is how the authors believe the legal status of sex work and stigma operate jointly – which is a central novel aspect of the present research. Addressing this issue could, in my opinion, significantly strengthen the potential theoretical contribution of the work. One possible position is that legality affects the amount of stigma experienced, and greater perceived and enacted stigma relate to greater risk for HIV (conceptually, mediation). Another position, one not necessarily incompatible, is that stigma will have a stronger relationship with HIV risk in countries in which sex work is fully criminalized vs. not specified vs. partially legalized (conceptually, moderated mediation). The rationale behind the hypotheses in the current work does not consider the processes at this level.

I strongly urge the authors need to develop their expectations and rationale more fully and precisely, because two conceptual models are directly relevant to the “how” and “why” questions that make the work more than simply descriptive. I agree with the authors that a basic limitation of the present work is that it is cross-sectional/correlation, which prevents causal inference, but I believe that more insights into the dynamics can and should be discussed. I think the authors data may suggest a double-impact of an illegal status of sex work, first in increasing stigma (potentially because it officially devalues the identity), and second in creating a stronger relationship between stigma and HIV risk because of the barriers it creates to seeking prevention or healthcare assistance (because of the cost associated with others becoming aware of the concealable identity.)

Another general reservation I have is about the cross-sectional/correlational nature of the design. While the cross-sectional design is a notable limitation, I do not believe it is a fatal flaw; the data the authors have are unique and speak to an important issue. However, I strongly recommend reframing the current work in a way that not only notes limitations of the cross-sectional design earlier but also expands conceptual consideration of possible ways legal status, stigma, and HIV risk might relate and how these processes could be reflected in the pattern of results obtained with these data. Although the authors are careful about using correlational rather than causal language when talking about their own data, much of the discussion strongly implies causal relations. For example, the authors write in the Discussion, “Our findings suggest that partial legalization, such as the removal of only some aspects of criminal laws and regulation of sex workers is necessary, but not sufficient for reducing sexual violence as a risk factor HIV,” which endorses a particular causal order. The authors own data, because it is correlational, could also support an argument of the opposite direction of causality: It may also be plausible that countries in which sex workers have a higher incidence of HIV might adopt stricter sex work laws. Some sort of time series analysis would be informative here. In its absence, I think the authors have to keep the cross-sectional design limitation more in readers’ consciousness as the authors form conclusions. Because of likely word-limit constraints, consideration of these dynamics could replace some of the things currently discussed and could be connected more closely to some of the recommendations posed. If, for example, the relationship between stigma and HIV risk is stronger where sex work is criminalized (presumably because seeking care puts people at legal risk), I am not sure that general empowerment interventions will be that effective.

In conclusion, I believe that the authors are addressing important theoretical and practical issues and that their data are valuable and their results are important. My reservation is that the current manuscript is under-theorized; I believe that the work would have much stronger impact – the type of

contribution that could merit publication in a journal with standards as high as this journal – if this aspect of the manuscript were developed further. These changes could be accomplished with some re-thinking, re-framing, and re-writing.

Reviewers' comments:

Reviewer #1 (Remarks to the Author):

General Comments:

The topic of this paper is timely and important, examining structural factors and HIV risk among female sex worker through seeking to understand the relationship between the legal environment regarding sex work, stigma, and HIV prevalence in sub-Saharan Africa. It draws on a unique and valuable cross-sectional survey data set with female sex workers across ten countries and leverages country-to-country variation in sex work laws to examine the relationship between HIV prevalence, stigma and criminalization of sex workers. Its major strength is that it consists of a large sample of a difficult-to-reach population---female sex workers across multiple countries where this is likely the only data of its kind. Its major weakness is the high risk of confounding across country contexts, most of which is not controlled for in the study.

Thank you and we sincerely appreciate this feedback and have aimed to address this overall comment as well as the specific comments below in the revised manuscript.

Specific comments:

Abstract

1. The abstract leads the reader to believe that there will be data presented on intersecting stigmas in the body of the text. To my understanding of intersectional stigma, none is presented in this paper, so I suggest removing the term from the abstract.

Thank you and we have removed this in the abstract and throughout the manuscript.

2. As this article is about female sex workers only, it would be good to qualify that in the abstract and throughout the text. (I realize it is in the title, but additional qualification throughout would help the reader)

Thank you and we have clarified this throughout the abstract and the text.

Introduction

3. I find the use of the word potentiate, both in the title and throughout the text odd and I think it may be difficult for most readers to grasp/understand. In looking up its meaning (I didn't know what this word meant), it seems to be used almost exclusively in the content of discussing increasing potency of drugs. Perhaps consider using a different word, like increasing, intensifying etc. to make the point of the paper clearer to the average reader, both in the title and in the text.

Thank you for this feedback and we have revised this to 'increasing'.

Methods

4. Were respondents asked about their HIV status first, before they were tested for HIV? Was pre and post-test counseling done before or after the questionnaire/stigma data collection? If respondents self-reported as HIV positive, was length of knowing HIV status collected? I ask because I wonder if sex workers who have been living with HIV for a while, and so may be experiencing intersectional stigma related to both HIV status and sex work, may be reporting higher levels of sex work stigma than those who are HIV negative and/or do not know their status. (which could imply intersectional stigma?) This might be worth discussing in the discussion section. What are the potential implications, particularly with respect to the relationship found between HIV status and stigma?

Thank you for highlighting this important aspect of potential intersectionality of stigmas due to attribution of sex work and/or HIV status. We have clarified our study procedures and as described in the text added below, we administered pre-test counseling and the blood draw prior to the questionnaire, however post-test counseling and delivery results were conducted following the questionnaire. Participants were administered an HIV test regardless of self-reported HIV status. We have provided more detail on study procedures in the methods section to clarify the process for biological testing alongside pre- and post-test counseling procedures as outlined below:

"Interviewer-administered socio-behavioral questionnaires were conducted and socio-behavioral measures were self-reported. All interviews were conducted in a private location with trained study staff. Biological testing for HIV, including pre- and post-test counseling, was conducted consistent with country-specific national guidelines. Participants with a reactive test result were referred to care. Pre-test counseling and biological testing were conducted prior to administering the socio-behavioral questionnaires. Post-test counseling and HIV test results were disclosed to the participants after completion of the socio-behavioral questionnaire."

In response to this comment we have now included the variable regarding previous knowledge of living with HIV, either from being told by a healthcare provider or receiving a positive HIV test result. Unfortunately, time from diagnosis was not collected from participants. The variable, knowledge of living with HIV, is now included in Table 3: Demographic characteristics, HIV risk and infection, disclosure, and stigma by legal status of sex work. We have also conducted a supplementary analysis to look at stigma by knowledge of HIV positive status, which is shown in Table Supplementary Table 2: Stigma by knowledge of living with HIV. This table aims to understand stigma exposures between those who were informed or aware of living with HIV prior to the study, compared to those who were either not living with HIV or unaware of their HIV positive status prior to the study. As suggested, knowledge of HIV status and stigma exposure are associated based on a Chi-squared test, with higher levels of stigma among those who were aware of living with HIV. All stigma measures, except for physical and sexual violence, reported experience of stigma

as being attributable to engagement in sex work. However, this shows that even stigma attributable to sex work is higher among those who are aware of their status as living with HIV, suggesting intersectionality.

We have added the following text to the result section: “Stigma exposure differs between participants with prior knowledge of living with HIV compared to those without prior knowledge of living with HIV or not living with HIV (Supplemental Table 2).”

Supplemental Table 2: Stigma and Knowledge of Living with HIV

		Knowledge of living with HIV among all participants				P value
		No		Yes		
Stigma		n/N	column %	n/N	column %	
Perceived	Family exclusion	542/4758	11.4	235/1194	19.7	<0.001
Perceived	Family gossip	917/4778	19.2	327/1198	27.3	<0.001
Perceived	Friend rejection	563/4731	11.9	258/1198	21.5	<0.001
Anticipated	Afraid of seeking health services	552/4807	11.5	200/1207	16.6	<0.001
Anticipated	Avoided seeking health services	400/4405	9.1	76/1029	7.4	0.083
Perceived	Mistreated in health center	110/4775	2.3	60/1207	5.0	<0.001
Enacted	Health care provider gossip	200/4776	4.2	82/1206	6.8	<0.001
Enacted	Denied health services	49/4809	1.0	28/1207	2.3	<0.001
Perceived	Police refused protection	611/4718	13.0	318/1202	26.5	<0.001
Perceived	Scared in public places	620/4511	13.7	161/963	16.7	0.017
Enacted	Verbally harassed	2206/4408	50.1	572/1030	55.5	0.002
Enacted	Blackmailed	1570/4808	32.7	415/1207	34.4	0.253
Enacted	Physical violence*	1545/4804	32.2	518/1205	43.0	<0.001
Enacted	Forced to have sex*	1473/4799	30.7	459/1200	38.3	<0.001

*Not specified as attributable to sex work

We have also included discussion on intersectional stigmas: “Female sex workers in this study may have also experienced intersecting stigmas attributable to both sex work and HIV status, as participants who reported to be aware of living with HIV prior to enrollment experienced higher levels of sex work-related stigmas. The combined or compounded effect of multiple stigmas may further influence uptake of services and health outcomes.¹⁻³ Leveraging innovative approaches to provide services outside of health facilities while working to mitigate observed individual and intersecting stigmas may facilitate improved service coverage.”

5. The sampling approach requires more detail. How was RDS conducted? How many seeds in how many locations? How long were the referral chains? How were the interviews conducted, etc?

Thank you and we have added more detail in the methods section to clarify the process for RDS, which is outlined below. We have also added details on interview procedures in response to the comment above. Lastly, we have updated Table 1: Summary of data collection to include the details for recruitment.

RDS, a peer-recruitment method designed to sample marginalized populations, was administered independently across the country-specific sites, to recruit female sex workers. Recruitment chains were initiated by seeds in each site, who were individuals selected in collaboration with local community-based organizations to represent heterogeneity in demographic characteristics and geographic representation. Initial seeds were provided with three coupons to recruit peers into the study. Women recruited by seeds and enrolled in the study were provided with three coupons for continued recruitment of peers. This process was repeated until reaching the target sample size of each country. Sample size calculations for the initial data collection were powered to estimate HIV prevalence at each site. The number of recruitment seeds by study site are provided in Table 1.

Table 1: Summary of data collection

Region	Country	Recruitment Dates	Country sample size	Study sites	Recruitment seeds	Total enrolled by site
West Africa	Burkina Faso	January – August 2013	699	Bobo Dioulasso	3	350
				Ouagadougou	6	348
	Senegal	February – November 2015	758	Dakar	9	502
				Mbour	3	256
	Côte d’Ivoire	March – October 2015	466	Abidjan	5	466
	Guinea-Bissau	September 2017 – January 2018	567	Bafatá	3	140
				Bissau	8	323
				Bissorã	3	45
				Gabu	3	59
	The Gambia	May 2017– May 2018	354	Banjul	9	354
Togo	January – June 2013	684	Kara	5	329	
			Lome	5	354	
Central Africa	Cameroon	November 2015 – October 2016	2255	Yaoundé	2	574
				Douala	1	457
				Bamenda	1	341
				Bertoua	1	304
				Kribi	1	579
Southern Africa	Lesotho	February - September 2014	744	Maseru	7	410
				Maputsoe	12	334
	Kingdom of eSwatini	August - October 2011	325	Manzini	14	324
	South Africa	October 2014 – April 2015	410	Port Elizabeth	9	410

6. While mentioned that this data was collected using RDS, it is unclear if it is being analyzed as RDS. It seems, that it’s being analyzed like a simple random sample (with some analyses being clustered by country). RDS sampling typically requires the application of RDS weights and some approach to account for observations’ non-independence in referral chains. If the analysis doesn’t incorporate weights accounting for clustering by referral chain (or another method of addressing standard errors – there’s a substantial debate in the field about how best to account for observations’ non-independence), it should be explained.

Thank you and we have clarified that crude estimates were used, instead of RDS-adjusted estimates. The analysis section now includes the following text: “Data were pooled and analyzed as crude data; RDS-adjusted weighting was not applied across countries as women did not represent a single network of female sex workers, violating a key assumption of RDS.⁴ Models were clustered by country and by site and represent valid sample estimates, but may not represent population-level estimates given the lack of RDS-adjustment.”

7. Were the data re-weighted when pooled? For example, Cameroon represents about 1/3 of the data and more than half of the data from fully criminalized settings. Is the Cameroon sample proportional to the estimated size of the population of sex workers in each setting? Are the results sensitive to how much weight is given to the observations from each location?

Thank you for this comment. The sample sizes for each study site/country are not proportional to the estimated size of the population of sex workers in each setting. Data in the analyses were not weighted when pooled for the final analyses presented in this manuscript. Based on the feedback regarding the size of the Cameroon sample and the reviewer’s helpful comment about sensitivity of the results to the weight given to the observations from each

location, we have conducted a sensitivity analysis using a random sample of the Cameroon dataset (n=700/2255). Using this smaller random sample from Cameroon pooled with the data from all other countries, we re-ran the analyses on legal status and compared findings to assess the potential influence of the large sample size from a country with criminalized status. We present these results in Supplementary Table 1. Overall, results remain similar in direction and magnitude to the analyses conducted using the full dataset. We have added discussion of these analyses to the manuscript and can make the results available as supplemental material.

Supplementary Table 1: HIV infection and country level legal status with random sample of Cameroon

	Living with HIV								
	n/N	%	X ² p value	OR	P value	95% CI	aOR*	P value	95% CI
Legal status of sex work			<0.001						
Partially legalized	219/1895	11.6		Ref	Ref	Ref	Ref	Ref	Ref
Selling not specified	248/1266	19.6		1.86	0.103	0.88,3.94	2.31	0.029	1.09,4.90
Criminalized	1225/2521	48.6		7.23	0.001	2.19,24.22	9.69	0.001	2.60,36.11

*Adjusted for age, education level, marital status, years in sex work, clustered by country and site

8. The association between each indicator of stigma and HIV is separately modeled rather than combined because of collinearity between the items. This indicates that they are potentially capturing the same underlying construct and I would also expect that many load onto the same factor, for example, family exclusion and family gossip to be indicators of the same underlying construct. Was any factor analysis done and if so, what is the rationale for individual items versus scaling?

Thank you and we appreciate this comment regarding the value of factor analysis for understanding the measurement of the constructs that may underlie these indicators. We performed an exploratory factor analysis to observe how items loaded onto different factors, and this roughly suggested 5 to 6 different factors to be the best fitting solution. We also observed heterogeneity across settings in the exploratory factor analysis. Given this large number of underlying factors, we decided to look at the relationship between key outcomes and individual items rather than scales, in order to better understand the granularity of these measures. We do agree that in the future studying latent constructs of stigma affecting sex workers and measurement invariance across countries is fundamentally important.

9. Curious about the background demographic variables that were collected. It seems only two were collected—age and education? Or were others collected and not included in the paper because they were non-significant? What was the rationale for including just these two? Was length in sex work collected? It seems that would could potentially be an important variable with respect to the experience of stigma—since stigma items are asked as ever happened questions? Marital status seems like potentially another important variable, along with disclosure of sex work status.

Thank you for this feedback and we added several additional variables with the aim of adjusting for background characteristics and potential confounders. We added marital status (currently married/not currently married); number of years engaging in sex (less than 5 years/five years or more); disclosure to family of engagement of sex work; and disclosure to healthcare provider of engagement of sex work. The multivariable models assessing stigma and sex work laws all now adjust for age, education, marital status, years in sex work. In addition to the set of covariates outlined above, the models assessing stigma relating to family or friends includes disclosure to family of engagement of sex work; and models assessing stigma related to the healthcare setting adjust for disclosure of engagement in sex work to a healthcare provider as based on prior literature these variables would be potential confounders for these specific analyses. Additionally, we included country level epidemic as a covariate and potential confounder in analyses assessing stigma and HIV (see response to comment 10 below).

Results

10. The associations between the legal status of sex work and HIV seem likely to be highly confounded. The same is true for associations between stigma and HIV stratified by legal status. Places where sex work is criminalized are mainly southern and central Africa; places where it is partially legalized or not specified are mainly west Africa. Without more theory and stronger control variables, one cannot know whether the observed associations are spurious and simply

reflect regional differences in HIV risk, whether greater HIV prevalence led to more criminalization, etc. I am not certain what the best way to account for underlying HIV risk is – perhaps a variable for local prevalence level?

Thank you for this helpful feedback. We have included a new variable in this analysis which categorizes the country level HIV epidemic as either generalized or concentrated epidemics. These categories leverage the traditional World Health Organization and UNAIDS definitions. Thus, a concentrated HIV epidemic includes countries in which HIV prevalence is consistently over 5% in at least one defined population, but less than 1% among reproductive aged women overall; a generalized HIV epidemic has an HIV prevalence consistently exceeding 1% in adult women. We examined the theoretical relationship of the country level epidemic with HIV infection, legal status of sex work, and stigma and included the epidemic variable in several sensitivity analyses which are discussed below.

Individual level HIV infection and country level HIV epidemic conceptually may be related in a bidirectional manner. The level of the HIV epidemic and the HIV prevalence among any specific population may influence the risk of exposure and potential transmission to an individual. Conversely, individual level infection contributes to the HIV prevalence within the population and therefore may influence the country level epidemic.

When assessing the potential relationships between country level HIV epidemic and legal status of sex work, we looked to the possible directionality of the HIV epidemic on sex work laws. Countries included in these analyses in which sex work is fully criminalized all established initial forms of criminalizing laws prior to the first detection of HIV infection globally. This eliminates the possibility that the establishment of criminalizing laws was a result of, or influenced by, the HIV epidemic. Based on these considerations, we assert that country level HIV epidemic conceptually would be a potential mediator of the relationship between sex work laws and HIV infection, rather than a confounder, and controlling for a mediator (country level HIV epidemic) may attenuate the relationship by removing one pathway by which sex work laws may have impacted HIV infection. We have conducted a sensitivity analysis to explore this relationship and provided the results in the table below (Response Table 1). These results show the directionality of the relationship between sex work laws and HIV remains the same when adjusted for the epidemic, and that the estimates attenuate towards the null but remain significant. These results are consistent with potential mediation as we hypothesize above. Given country epidemic would not be considered a potential confounder and instead appears to be a potential mediator, we did not adjust for it in the final model assessing the relationship between legal status of sex work and HIV. We have incorporated this justification into the methods section. We have not included Response Table 1 results in the supplementary material but are happy to do this if this is recommended.

Response Table 1: HIV infection and country level legal status

	Living with HIV									Living with HIV adjusted for country level epidemic		
	n/N	%	X ² p value	OR	P value	95% CI	aOR*	P value	95% CI	aOR**	P value	95% CI
Legal status of sex work			<0.001									
Partially legalized	219/1894	11.6		Ref	Ref	Ref	Ref	Ref	Ref	Ref	Ref	Ref
Selling not specified	248/1265	19.6		1.87	0.181	0.78,4.65	2.35	0.036	1.05,5.21	2.52	0.025	1.12, 5.68
Criminalized	1603/4071	39.4		4.97	0.001	1.98,12.44	7.17	<0.001	2.71,19.00	2.71	0.028	1.28, 5.73

*Adjusted for age, education level, marital status, years in sex work, clustered by country and site

** Adjusted for age, education level, marital status, years in sex work, country level HIV epidemic clustered by country and site

After considering the potential role of confounding of the country level HIV epidemic on the relationship between legal status of sex work and the outcome of HIV prevalence (described above), we next considered the potential role of confounding of the country level HIV epidemic on the relationship between stigma and HIV prevalence. We hypothesize that stigma and country level epidemic may have a bidirectional relationship. That is, the country level epidemic may influence stigma associated with HIV (for example, in lower prevalence settings stigma may be higher as fewer individuals know someone living with HIV and may associate HIV with individuals engaged in behaviors which increase risk, such as sex work). Conversely, stigma may influence the HIV epidemic through limited uptake and provision of quality HIV services, potentially increasing new infections. Therefore, we agree with the reviewer that country level epidemic may be a potential confounder in the relationship between stigma and HIV infection. Given

this, we have now included country level epidemic in the analyses assessing the relationship between stigma and HIV. Results are included in updated manuscript and tables.

11. The customary way to assess relationships between legal status and health outcomes is to use a difference-in-differences/controlled before-after approach or, if not possible, before-after with careful statistical controls. Comparing cross-sectional data from different legal contexts is fraught because there is a high risk of confounding and reverse causation. At minimum, it requires discussion of a strong theoretical framework operationalized into control variables. (See, for example, Wagenaar's book and other materials at <http://publichealthlawresearch.org/theory-methods>.)

We agree with this perspective and appreciate this feedback on the methods for assessment. Ideally, we would have longitudinal data to more accurately assess the direction of this relationship and use a more customary method for this analysis. Although we are not able to assess causality through cross-sectional data, as described above, we do leverage the time periods in which laws were established to provide some interpretation and context to these relationships. Countries in our analyses in which sex work is fully criminalized all established initial forms of criminalizing laws prior to the first detection of HIV infection globally. This fact at a minimum rules out the possibility of reverse causation that the establishment of criminalizing laws were a result or influenced by the HIV epidemic. We have added this to the methods and as a discussion point to the limitations section.

Thank you for sharing these references and tools and we have provided references and discussion of a theoretical framework to inform our analyses. We have included references to theoretical frameworks which guided the analyses stating: "Statistical models were guided by the Structural HIV Determinants Framework for Sex Work and the Logic Model of Public Health Law Research."⁵

12. Please cross-check all the figures in the text with the tables—there are several that are different in the text compared to the tables. For example, reported verbal harassment 31.2% (2200/7252) in the text and 30.34% in table 3. There are others as well.

Thank you and we have corrected this.

13. Given the detail is all in the tables, would suggest not repeating every single figure/result in the text, but instead summarizing or highlighting key trends or differences from the tables.

Thank you and we have reduced narrative in the results, and instead focused on key trends or differences.

14. To help the reader, I would suggest putting the relevant table(s) for each section next to the header for each section, e.g. HIV and stigma (table 3 and 4); The results section is quite dense and this will help the reader follow along more easily, to have the referent table listed up at the start of each section

Thank you and we have added this, and also aimed to simplify the results section. In reviewing the manuscript guidelines this does not seem to be the standard for the journal, although we are happy to include or exclude based on the preference.

15. Legal status of sex work section (starts at line 160)—which is table 5---also includes association of legal status with stigma (table 6)—essentially the last line in this section only. Suggest modifying the header to say Legal status of sex work (Table 5) and stigma (table 6)

Thank you and we have revised this.

16. Line 168/169. Stigma was associated with legal status of sex work except for family gossip and denial of healthcare (table 6). Table 6 indicates that family gossip was significant, but friend rejection was not. Which is correct—what is in the text or what is in the table?

Thank you and we have corrected this.

17. HIV and stigma by legal status of sex work (Table 7). In presenting the results of the regression analysis, it will be important to note that in some cases the odds ratios are below 1—indicating that the relationship between the stigma item and HIV infection is significantly lower for the selling not specified setting versus the comparison legal setting—partially legalized. (family gossip, scared in public places and physical violence). Based on the rest of the text of the paper, I'm assuming this is an unexpected finding (that one would expect the odds to be higher in the selling not specified than partially legalized) or maybe it is not (in which case where the odds are higher would need discussion). Some discussion of the fact that these relationships change for different stigma items and the partially legalized vs selling not specified in the discussion section is warranted.

Thank you and we have added this to the discussion. With the revised results, fewer stigma measures showed a negative association with HIV infection, however we discuss this for fear of public places in the discussion, which is included: "In this study, fear of being in public places was negatively associated with HIV prevalence overall, prior to stratification across legal contexts. In part, the lower HIV prevalence may emerge from protective behaviors such as avoiding street-based sex work which has generally been associated with increased violence, extortion by uniformed officers, and increased HIV-related risk behaviors. ^{6,7}"

Discussion

18. I'm not sure it's customary to call this kind of study a meta-analysis, since so far as I can tell the data haven't been published previously. Rather, it's really a multi-site survey, unless I'm missing something. Consider removing the term meta-analysis throughout the discussion section.

Thank you and we have removed any reference to meta-analysis.

19. See comment #4 above.

We have added the following to the limitations section: This study did not assess the potential intersections of stigmas attributable to sex work and HIV status.

20. See comment #17 above.

Thank you and we have added this to the discussion as outlined in response to #17.

Limitations

21. The studies were conducted over a period of about seven years (which is reasonable considering the difficulty achieving large sample sizes in hard-to-sample populations). However, it would be worth discussing in the limitations section whether this poses any risk of confounding or bias. This would be particularly problematic if there are any systematic relationships expected between calendar time and legal status, stigma, or HIV risk.

Thank you and we have added the following statement to the limitations section: "Data were collected over a period of seven years, which should be considered in the interpretation of the results. Enforcement practices, program funding, and other external measures over time may have influenced stigma, HIV status, or HIV risk."

Reviewer #2 (Remarks to the Author):

The issue that the authors address, the relationship of stigma and sex work laws with HIV risks among female sex workers, is timely and socially important. The data, collected by the research team across 10 sub-Saharan countries, is unique and valuable. The examination of individual-level data stratified by country-level legal status of sex work represents a sophisticated analytical approach. The results are generally strong and consistent, and supportive of expectations. These aspects of the work make it a potential candidate for publication.

However, despite my generally favorable orientation to this work because of its practical implications, I do have a

number of reservations about conceptual aspects of the research. One reservation that I have concerns the magnitude of new conceptual contribution of the research. What makes this research unique is not so much the findings of a relationship between stigma and HIV risk – a number of studies of various groups already document such a relationship – but the particular focus of the current work on female sex workers and differences related to the legal status of sex work. As the authors note, a very small percentage of studies in the literature concern stigma associated with sex work. However, as the authors acknowledge, there is other work also linking laws about sex work and incidence of HIV: “Mathematical modelling suggests that across generalized and concentrated HIV epidemics, decriminalization of sex work could have the largest effect on the course of the HIV epidemic, averting one third to almost one half of incident HIV infections over the next decade.” Nevertheless, I do think that the empirical evidence presented by the authors may be sufficiently unique for publication – if the authors make the nature of new contribution more conceptually compelling.

Thank you. We understand and agree with the reviewer’s concern that the novelty of these analyses and the importance of individual level, empiric data to answer the underlying questions could be presented in a way that more clearly delineates the contribution of this work to the field. We have made substantial edits to the introduction and the discussion to address this concern.

Additionally, as a small adjustment, we have reframed our descriptive table (Table 2) to focus around characteristics across the different legal categories, again trying to re-enforce the focus around the legal status of sex work and the novel elements of this work. Based on this comment we felt this may better structure the aims and subsequent analyses.

I do not believe that the current manuscript makes a strong enough conceptual case, though. My concerns about this facet of the manuscript involve both the introductory framing of and interpretation of the work. The current introduction is limited in explaining how and why stigma increases HIV risk. The authors state that stigma “results in a separation from the social norms and leads to adverse experiences and a loss of social status.” While stigma is a devaluation of a person with a marked identity, people cope with stigma in a wide range of ways – not all of which would lead to greater HIV risk. Greater stigma, for example, could lead people to different work, which might lower HIV risk. Women who responded this way are not in the current sample. In addition, the authors make a distinction between enacted and perceived stigma in their measures. This is a common and important distinction. However, that distinction is only noted in passing, not really considered analytically, in the data analysis and interpretation.

Thank you and we have worked to reframe the introduction and objective of this work as well as clarify the interpretation throughout the discussion to improve the conceptual case of this analysis.

Thank you also for raising the point of needing to further address and discuss the different types of stigma in this analysis. While in our methodological approach, we treated all stigma measures separately and therefore did not combine items into scales based on stigma type (enacted, perceived, anticipated) or sector (family, health, or community), we appreciate the reviewer’s point that it is important to consider these different forms of stigma in interpretation. We have therefore highlighted the types of stigma throughout the discussion section.

We also considered the comment on how stigma may lower HIV risk in our interpretation of negative stigma association with HIV risk. For example the following was incorporated into the discussion: “In this study, fear of being in public places was negatively associated with HIV prevalence overall, prior to stratification across legal contexts. In part, the lower HIV prevalence may emerge from protective behaviors such as avoiding street-based sex work which has generally been associated with increased violence, extortion by uniformed officers, and increased HIV-related risk behaviors.”

Another key point that I believe needs to be developed further is how the authors believe the legal status of sex work and stigma operate jointly – which is a central novel aspect of the present research. Addressing this issue could, in my opinion, significantly strengthen the potential theoretical contribution of the work. One possible position is that legality affects the amount of stigma experienced, and greater perceived and enacted stigma relate to greater risk for HIV (conceptually, mediation). Another position, one not necessarily incompatible, is that stigma will have a stronger

relationship with HIV risk in countries in which sex work is fully criminalized vs. not specified vs. partially legalized (conceptually, moderated mediation). The rationale behind the hypotheses in the current work does not consider the processes at this level.

Thank you and we have revised the manuscript throughout to have this come across more clearly. In the introduction we have aimed to more clearly reference the driving hypothesis, in that stigma and laws act jointly in increasing HIV risk and that the relationship between stigma and HIV is strongest in criminalized and non-specified settings.

I strongly urge the authors need to develop their expectations and rationale more fully and precisely, because two conceptual models are directly relevant to the “how” and “why” questions that make the work more than simply descriptive. I agree with the authors that a basic limitation of the present work is that it is cross-sectional/correlation, which prevents causal inference, but I believe that more insights into the dynamics can and should be discussed. I think the authors data may suggest a double-impact of an illegal status of sex work, first in increasing stigma (potentially because it officially devalues the identity), and second in creating a stronger relationship between stigma and HIV risk because of the barriers it creates to seeking prevention or healthcare assistance (because of the cost associated with others becoming aware of the concealable identity.)

Thank you and we have revised the manuscript throughout to reinforce the suggested *how* and *why* behind the relationships observed. In both the introduction and the discussion we now speak to the hypothesized dynamics at play and the potential for impact of the illegality of sex work. For example we have added the following paragraph to the discussion: “Although stigma is prevalent and associated with legal status of sex work, reporting any history of stigma is not clearly or consistently higher in criminalized or non-protective settings compared to partially legalized settings. However, the relationship between stigma and HIV varies across different legal contexts of sex work, suggesting that stigma and sex work laws interact in increasing HIV risk. Sex workers living in settings with criminalized and non-specified laws mostly show a stronger relationship between stigma and HIV compared to partially legalized settings. Given the near universality of stigma affecting sex work, mechanisms increasing HIV risks in more punitive settings include challenges in the provision as well as the uptake of HIV prevention and treatment services and differential levels of social capital and resiliency among sex workers. These findings suggest that sex workers in punitive and non-protective environments may be more susceptible to harms and HIV risks related to stigma. Sex workers in partially legalized or more protective environments may have higher levels of resilience that can mitigate the impact of stigma on HIV risks.(38) Ultimately, these results suggest the potential impact of stigma on HIV risk may be greatest in punitive and non-protective settings, and that interventions aiming to address stigma and HIV risks may be more effective if implemented with consideration of the legal context.”

We have also tried to more clearly address the potential “how” and “why” in each of the discussion paragraphs focused on healthcare related stigma, violence, and lack of protection from police.

Another general reservation I have is about the cross-sectional/correlational nature of the design. While the cross-sectional design is a notable limitation, I do not believe it is a fatal flaw; the data the authors have are unique and speak to an important issue. However, I strongly recommend reframing the current work in a way that not only notes limitations of the cross-sectional design earlier but also expands conceptual consideration of possible ways legal status, stigma, and HIV risk might relate and how these processes could be reflected in the pattern of results obtained with these data. Although the authors are careful about using correlational rather than causal language when talking about their own data, much of the discussion strongly implies causal relations. For example, the authors write in the Discussion, “Our findings suggest that partial legalization, such as the removal of only some aspects of criminal laws and regulation of sex workers is necessary, but not sufficient for reducing sexual violence as a risk factor HIV,” which endorses a particular causal order. The authors own data, because it is correlational, could also support an argument of the opposite direction of causality: It may also be plausible that countries in which sex workers have a higher incidence of HIV might adopt stricter sex work laws. Some sort of time series analysis would be informative here. In its absence, I think the authors have to keep the cross-sectional design limitation more in readers’ consciousness as the authors form conclusions. Because of likely word-limit constraints, consideration of these dynamics could replace some of the things currently discussed and could be connected more closely to some of the recommendations posed. If, for example, the relationship between stigma and HIV risk is stronger where sex work is criminalized (presumably because seeking care

puts people at legal risk), I am not sure that general empowerment interventions will be that effective.

Thank you for this thoughtful feedback. In relation to your comment about directionality relating to sex work laws, we have added to the methods and discussion addressing the interpretation of our cross-sectional data. Although we are not able to assess causality through cross-sectional data, the introduction of sex work laws preceded the HIV epidemic expansion in each of these settings allowing some interpretation and context to these relationships. Thus, we can at a very minimum rule out the theoretical reverse causation that the establishment of criminalizing laws were a result or influenced by the HIV epidemic.

Specifically, we have added the following to the limitations section: “Although we are not able to assess causality through cross-sectional data, and cannot account for the relationship between HIV prevalence and stigma over time, laws were established prior to HIV introduction within countries. This at a minimum rules out the possibility that laws criminalizing sex work were a result of or influenced by the HIV epidemic. “

We have added the following to the Methods section: “Although the country-level epidemic (concentrated vs. generalized) is associated with HIV prevalence, it is not considered as a confounder in our conceptual model, but rather a mediator between sex work law and HIV prevalence, as sex work laws in each country preceded the introduction of HIV within the countries.”

Furthermore, we have also aimed to keep the cross-sectional design limitation more in readers’ consciousness in our formation of the conclusions. Furthermore, incorporating the recommended analytical adjustments from Reviewer 1 influenced the results relating to social stigma. Therefore, we felt it was appropriate to remove the discussion on empowerment interventions.

In conclusion, I believe that the authors are addressing important theoretical and practical issues and that their data are valuable and their results are important. My reservation is that the current manuscript is **under-theorized**; I believe that the work would have much stronger impact – the type of contribution that could merit publication in a journal with standards as high as this journal – if this aspect of the manuscript were developed further. These changes could be accomplished with some re-thinking, re-framing, and re-writing.

We thank the reviewer for these constructive and thoughtful comments and for supporting us in the strengthening of the framing and messaging related to this work. We believe that the revised manuscript is much stronger as a result of the feedback received.

References

1. Logie CH, Williams CC, Wang Y, et al. Adapting stigma mechanism frameworks to explore complex pathways between intersectional stigma and HIV-related health outcomes among women living with HIV in Canada. *Social science & medicine* (1982). 2019;232:129-138.
2. Turan B, Hatcher AM, Weiser SD, Johnson MO, Rice WS, Turan JM. Framing Mechanisms Linking HIV-Related Stigma, Adherence to Treatment, and Health Outcomes. *American journal of public health*. 2017;107(6):863-869.
3. Earnshaw VA, Chaudoir SR. From conceptualizing to measuring HIV stigma: a review of HIV stigma mechanism measures. *AIDS and behavior*. 2009;13(6):1160-1177.
4. Salganik MJH, Douglas D. . Sampling and Estimation in Hidden Populations Using Respondent-Driven Sampling. *Sociological Methodology*. 2004;34:193-239.
5. Burris S, Wagenaar AC, Swanson J, Ibrahim JK, Wood J, Mello MM. Making the case for laws that improve health: a framework for public health law research. *The Milbank quarterly*. 2010;88(2):169-210.
6. Deering KN, Lyons T, Feng CX, et al. Client demands for unsafe sex: the socioeconomic risk environment for HIV among street and off-street sex workers. *Journal of acquired immune deficiency syndromes (1999)*. 2013;63(4):522-531.

7. Platt L, Grenfell P, Meiksin R, et al. Associations between sex work laws and sex workers' health: A systematic review and meta-analysis of quantitative and qualitative studies. *PLoS medicine*. 2018;15(12):e1002680.

Reviewers' Comments:

Reviewer #1:

Remarks to the Author:

The authors have adequately addressed my concerns.

A couple minor issues to strengthen the paper further:

I am puzzled by finding the discussion section (starting on page 6) placed ahead of the methods (starting page 11).

The manuscript would benefit from a close copy edit.

Take a closer look at the discussion section and consider tightening it up and making it more succinct.

Reviewer #2:

Remarks to the Author:

I was Reviewer #2 on the original submission of this manuscript. I was very impressed by a number of aspects of the work – the unique and important dataset; the sophisticated multi-level conceptual and statistical analysis; the generally, strong and supportive results – but I also expressed some reservations about the magnitude of the theoretical contribution of the work, as well as a number of specific points of concern. I have carefully reviewed the revised manuscript and read and considered the responses to the comments the other reviewer and I made concerning the previous version of the work.

Overall, I believe that the authors have been very responsive to the guidance offered, and I feel this version is much improved. While I do think that this version merits my support for publication, I do have a couple of comments – lingering concerns – that I believe would still be valuable to address.

The ways that the authors have reframed the introduction and the clearer articulation of the novel objectives of the work make a more persuasive case for the value of the project. However, while the authors now explain more effectively their interests in “the mechanisms through which sex work laws and stigma potentiate HIV risks among female sex workers independent of individual-level behaviors” (p. 4), it is not clear what they mean by “mechanisms” and how these mechanisms are operationalized in the present research. I think they might be referring to the effects on variables associated elements of the HIV treatment cascade, but it would be useful to make a more explicit connection at the very end of the Introduction to the mechanisms the authors say they are studying with respect to specific engagement in prevention, care, and treatment services in the HIV treatment cascade. If these are not the mechanisms the authors intend, then the authors should directly identify the ones they mean.

The other concern that I continue to have in some degree involves the need for a bit more consideration of WHY the authors believe that there would be “generally stronger associations between stigma and HIV in punitive and non-protective settings compared to partially legalized settings” (p. 4). There is research (e.g., by Hatzenbuehler and others) about how laws in the US affect the feelings of stigma. But, this work would suggest that the effect on HIV would be due to less stigma experienced by sex workers partially legalized settings than in punitive and non-protective settings. The authors, however, argue that the level of stigma is not what is influential but rather it is that the impact of a given level of stigma is stronger in punitive and non-protective settings compared to

partially legalized settings. I am not saying that the authors have to switch their expectation (particularly since they report that the level of stigma experienced is not higher in punitive and non-protective settings, see p. 7), but I do think it would be valuable for them to say, rooted in conceptually and empirically in previous work, why they suggest the process that they do. The explanation that the authors present in the Discussion – that “sex workers in punitive and non-protective environments may be more susceptible to harms and HIV related risks related to stigma” (p. 7) – is more a restatement of the results than a theoretically stimulating interpretation of the mechanisms and processes that produce this effect. I commend the authors for emphasizing more the cross-sectional nature of their data and the consequent limitations for causal claims. However, even though they cannot definitively test the causal links with the present design, the idea that punitive and non-protective laws magnifies the impact of stigma (potentially without changing the level of stigma experienced) requires more substantiation, particularly because it is a key element of the take-home message (see last paragraph of p. 6) the authors offer to readers. My simple recommendation would be to suggest in the Introduction that the joint effect of laws and stigma could take the form of (a) producing the experience of greater stigma in punitive and non-protective settings compared to partially legalized settings because the former policies suggest that the characteristic is more deviant and less socially acceptable, (b) increasing the strength of the relationship between stigma and HIV risk because of the barriers created for seeking prevention or healthcare assistance, elements of the HIV treatment cascade, because of the potential repercussions of others becoming aware of their concealable identity, or (c) a combination of both. The Discussion could then make the case, with caution about the limitations of cross-sectional data for causal inference, that the second explanation is most plausible given the pattern of results.

In conclusion, I found this manuscript more compelling than the previous version. I believe that this is important work that I would like to see published in this journal. I still think some additional revision around a couple of points, which I see as fundamentally relating to the conceptual message of the research, would be important. However, despite the amount that I wrote about these in this review, there are some direct, “quick fixes” that I offer to alleviate the concerns. I hope the editor and authors find these comments helpful.

RESPONSE TO REVIEWERS

REVIEWERS' COMMENTS:

We sincerely appreciate the review and continued feedback from the reviewers. We have attempted to address the final comments in the revised manuscript.

Reviewer #1 (Remarks to the Author):

The authors have adequately addressed my concerns.

Thank you and we sincerely appreciated the thoughtful feedback and glad this has been adequately addressed.

A couple minor issues to strengthen the paper further:

I am puzzled by finding the discussion section (starting on page 6) placed ahead of the methods (starting page 11).

Thank you and per the editor's request, we have kept the order of the sections as is.

The manuscript would benefit from a close copy edit.

Thank you and we have conducted a close copy edit.

Take a closer look at the discussion section and consider tightening it up and making it more succinct.

Thank you and we have tried to tighten up the discussion.

Reviewer #2 (Remarks to the Author):

I was Reviewer #2 on the original submission of this manuscript. I was very impressed by a number of aspects of the work – the unique and important dataset; the sophisticated multi-level conceptual and statistical analysis; the generally, strong and supportive results – but I also expressed some reservations about the magnitude of the theoretical contribution of the work, as well as a number of specific points of concern. I have carefully reviewed the revised manuscript and read and considered the responses to the comments the other reviewer and I made concerning the previous version of the work.

Overall, I believe that the authors have been very responsive to the guidance offered, and I feel this version is much improved. While I do think that this version merits my support for publication, I do have a couple of comments – lingering concerns – that I believe would still be valuable to address.

Thank you and we have sincerely appreciated the thoughtful and constructive feedback you have provided. We have attempted to address the lingering concerns and remain very happy to make further revisions as needed.

The ways that the authors have reframed the introduction and the clearer articulation of the novel objectives of the work make a more persuasive case for the value of the project. However, while the authors now explain more effectively their interests in “the mechanisms through which sex work laws and stigma potentiate HIV risks among female sex workers independent of individual-level behaviors” (p. 4), it is not clear what they mean by “mechanisms” and how these mechanisms are operationalized

in the present research. I think they might be referring to the effects on variables associated elements of the HIV treatment cascade, but it would be useful to make a more explicit connection at the very end of the Introduction to the mechanisms the authors say they are studying with respect to specific engagement in prevention, care, and treatment services in the HIV treatment cascade. If these are not the mechanisms the authors intend, then the authors should directly identify the ones they mean.

Thank you and we really appreciate this feedback and have tried to more clearly describe this in the introduction. Throughout the paper, we have attempted to clarify that the mechanisms through which sex work laws and stigma potentiate HIV risks among female sex workers act to create barriers to safety and access to services, but that the specific mechanisms depend on the type of stigma. For example, we expect that the specific mechanism for how stigma increases HIV risk for police refusal to provide protection or violence will be different from mistreatment in the healthcare setting. We have tried to make this come across more clearly, while respecting the word limit provided. Despite the brevity in the introduction due to available space, we have tried to provide this detail in the discussion section.

Below, please find the updated study description in the introduction:

In response, this study aims to use individual-level data to characterize the relationship between sex work laws, stigmas, and HIV risks among female sex workers across sub-Saharan Africa. Collectively, findings from these analyses suggest that increasingly punitive and non-protective laws are associated with increased odds of prevalent HIV infection among sex workers. Furthermore, stigma and sex work laws may operate synergistically in increasing HIV risks, with generally stronger associations between stigma and HIV in punitive and non-protective settings compared to partially legalized settings. The results suggest that the increased harmful effects of stigma in more punitive and non-protective legal contexts may be due increased barriers in the provision or uptake of efficacious HIV prevention and treatment services, or impunity among perpetrators of stigma and lack of recourse for sex workers experiencing health and social stigmas, or more likely, a combination of the two.

The other concern that I continue to have in some degree involves the need for a bit more consideration of WHY the authors believe that there would be “generally stronger associations between stigma and HIV in punitive and non-protective settings compared to partially legalized settings” (p. 4). There is research (e.g., by Hatzenbuehler and others) about how laws in the US affect the feelings of stigma. But, this work would suggest that the effect on HIV would be due to less stigma experienced by sex workers partially legalized settings than in punitive and non-protective settings. The authors, however, argue that the level of stigma is not what is influential but rather it is that the impact of a given level of stigma is stronger in punitive and non-protective settings compared to partially legalized settings. I am not saying that the authors have to switch their expectation (particularly since they report that the level of stigma experienced is not higher in punitive and non-protective settings, see p. 7), but I do think it would be valuable for them to say, rooted in conceptually and empirically in previous work, why they suggest the process that they do. The explanation that the authors present in the Discussion – that “sex workers in punitive and non-protective environments may be more susceptible to harms and HIV related risks related to stigma” (p. 7) – is more a restatement of the results than a theoretically stimulating interpretation of the mechanisms and processes that produce this effect.

Thank you for this helpful feedback. We have restructured the discussion in response to this comment, hoping to better describe the ‘why’.

Please find the updated section focused on this below:

The relationship between stigmas and HIV varies across different legal contexts of sex work, suggesting that stigmas and sex work laws interact in increasing HIV risks and ultimately burden. Sex workers living in settings with criminalized and non-specified laws generally show a stronger relationship between stigmas and HIV burden compared to partially legalized settings. Existing evidence suggests that sex workers living in punitive and non-protective settings may experience greater burden of stigmas than women living in partially legalized settings. However, in this study women reporting any lifetime history of stigmas is not clearly or consistently higher in criminalized or non-protective settings compared to partially legalized settings, highlighting that sex workers across legal environments experience stigmas. While sex workers may still experience a greater frequency of stigmas over the course of their lifetime in punitive and non-protective settings, the periodicity of stigma experiences among women is not measured in this study. Given the near universality of stigmas affecting sex workers, the mechanisms associated with increased HIV burden may act by amplifying the barriers to safety, as well as efficacious health services. Specifically, sex workers in punitive and non-protective environments may be more susceptible to the harms related to stigmas affecting overall safety in society and in access to HIV prevention and treatment services. Furthermore, sex workers in partially legalized or more protective environments have been shown to have higher levels of social capital, resiliency, and options for support that can mitigate the impact of stigmas on HIV risks. Ultimately, the mechanisms underpinning the synergies of stigma and sex work laws in the burden of HIV among sex workers likely vary by the specific type of stigma. The consistency in the findings of the interaction between laws and stigma in especially punitive legal settings reinforce the importance of HIV prevention and treatment intervention strategies tailored for sex workers that consider the legal context during implementation.

I commend the authors for emphasizing more the cross-sectional nature of their data and the consequent limitations for causal claims. However, even though they cannot definitively test the causal links with the present design, the idea that punitive and non-protective laws magnifies the impact of stigma (potentially without changing the level of stigma experienced) requires more substantiation, particularly because it is a key element of the take-home message (see last paragraph of p. 6) the authors offer to readers. My simple recommendation would be to suggest in the Introduction that the joint effect of laws and stigma could take the form of (a) producing the experience of greater stigma in punitive and non-protective settings compared to partially legalized settings because the former policies suggest that the characteristic is more deviant and less socially acceptable, (b) increasing the strength of the relationship between stigma and HIV risk because of the barriers created for seeking prevention or healthcare assistance, elements of the HIV treatment cascade, because of the potential repercussions of others becoming aware of their concealable identity, or (c) a combination of both. The Discussion could then make the case, with caution about the limitations of cross-sectional data for causal inference, that the second explanation is most plausible given the pattern of results.

Thank you and in the introduction we include the following: Laws criminalizing sex work may both increase sex work-related stigma and potentially contribute to epidemic growth if the sequelae of these laws increases vulnerability and decreases engagement in HIV prevention and treatment services.

Thank you and per this recommendation, we now include the following in the introduction: The results suggest that the increased harmful effects of stigma in more punitive and non-protective legal contexts may be due increased barriers in the provision or uptake of efficacious HIV prevention and

treatment services, or impunity among perpetrators of stigma and lack of recourse for sex workers experiencing health and social stigmas, or more likely, a combination of the two.

We have now clarified in the discussion that although we did not observe clear differences in the lifetime prevalence of stigma (any/none) in criminalized or non-protective settings compared to partially legalized settings, it may be that the frequency of stigma experienced by each individual may differ by legal context. Unfortunately, our measures of stigma are ever/never lifetime exposures of stigma, however we do think that is likely that sex workers in punitive and non-protective settings may experience stigma more frequently than women in partially legalized settings. This difference in frequency may partially explain the varying strength of the relationship of stigma on HIV across settings.

In conclusion, I found this manuscript more compelling than the previous version. I believe that this is important work that I would like to see published in this journal. I still think some additional revision around a couple of points, which I see as fundamentally relating to the conceptual message of the research, would be important. However, despite the amount that I wrote about these in this review, there are some direct, “quick fixes” that I offer to alleviate the concerns. I hope the editor and authors find these comments helpful.

Thank you and we sincerely appreciate this constructive and thoughtful feedback.